# Rethinking Unsupervised Cross-modal Flow Estimation: Learning from Decoupled Optimization and Consistency Constraint

**Runmin Zhang**[1]   **Jialiang Wang**[1]   **Si-Yuan Cao**[2,3,4*]   **Zhu Yu**[1]
**Junchen Yu**[1]   **Guangyi Zhang**[1]   **Hui-Liang Shen**[1]

[1]College of Information Science and Electronic Engineering, Zhejiang University
[2]Ningbo Global Innovation Center, Zhejiang University   [3]NingboTech University
[4]Jinhua Institute of Zhejiang University
{runmin_zhang, cao_siyuan}@zju.edu.cn

## Abstract

This work presents DCFlow, a novel self-supervised cross-modal flow estimation framework that integrates a decoupled optimization strategy and a cross-modal consistency constraint. Unlike previous unsupervised approaches that implicitly learn flow estimation solely from appearance similarity, we introduce a decoupled optimization strategy with task-specific supervision to address modality discrepancy and geometric misalignment distinctly. This is achieved by collaboratively training a modality transfer network and a flow estimation network. To enable reliable motion supervision without ground-truth flow, we propose a geometry-aware data synthesis pipeline combined with an outlier-robust loss. Additionally, we introduce a cross-modal consistency constraint to jointly optimize both networks, significantly improving flow prediction accuracy. For evaluation, we construct a comprehensive cross-modal flow benchmark by repurposing public datasets. Experimental results demonstrate that DCFlow can be integrated with various flow estimation networks and achieves state-of-the-art performance among unsupervised approaches. The source code is available at https://github.com/RM-Zhang/DCFlow.

## 1 Introduction

Cross-modal flow estimation aims to establish pixel-wise correspondences between images captured from different modalities. It is crucial for various vision tasks, including multi-modal image fusion (Liu et al., 2025), image restoration (Zhang et al., 2025), and depth estimation (Guo et al., 2023). Due to the difficulty of acquiring cross-modal ground-truth flow in real-world scenarios, unsupervised cross-modal flow estimation, which does not require such annotations for training, has attracted increasing attention.

Existing unsupervised approaches typically address this task by minimizing appearance discrepancies between image pairs. To achieve this, a modality transfer network is usually employed to translate images from one modality to another. For instance, NeMAR (Arar et al., 2020) simultaneously optimizes the modality transfer and flow estimation networks, while others (Wang et al., 2022; Xu et al., 2022) adopt a two-stage pipeline for separate training. Despite their different strategies, these methods share a fundamental limitation in implicitly learning flow estimation solely through appearance alignment. Consequently, they struggle particularly in textureless regions or repetitive structures, and their performance substantially degrades under large viewpoint changes due to the lack of direct flow supervision. This naturally raises a question: can we introduce reliable flow supervision using only unaligned cross-modal image pairs? Recent studies (Watson et al., 2020; Aleotti et al., 2021; Han et al., 2022; Liang et al., 2023) have explored generating synthetic motion labels from single images via geometry-aware data synthesis, showing promising results in mono-

---

*Corresponding author.

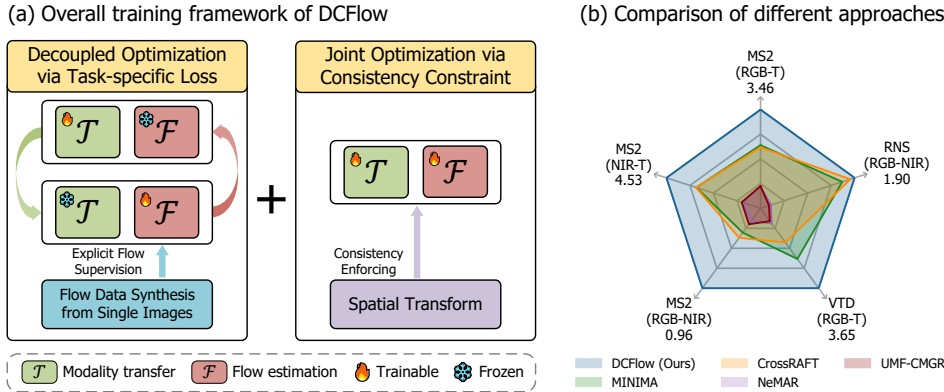

Figure 1: (a) Overall training framework of DCFlow, which integrates a decoupled optimization strategy and a cross-modal consistency constraint. (b) Comparison of the cross-modal flow estimation accuracy on five datasets. EPEs (endpoint errors) of different approaches are reported. Our DCFlow achieves state-of-the-art performance.

modal settings. However, whether such mono-modal supervision can benefit cross-modal scenarios, and how to effectively exploit it, remain largely unexplored.

To bridge this gap, in this paper, we propose DCFlow, a novel self-supervised framework for cross-modal flow estimation that introduces explicit motion supervision into the training process. In line with existing approaches, our framework consists of a modality transfer network and a flow estimation network. As shown in Fig. 1(a), DCFlow integrates a decoupled optimization strategy with task-specific supervision to train the two networks separately, as well as a cross-modal consistency constraint to jointly optimize them. Within this framework, we leverage complementary sources of motion supervision through geometry-aware synthesis from single images and spatial transformations on cross-modal pairs. In this way, DCFlow learns in a fully self-supervised manner by generating flow supervision through data synthesis, using only unaligned cross-modal images.

The decoupled optimization strategy separates the overall task into modality transfer and single-modal flow estimation, forming a collaborative process where each network facilitates the optimization of the other. More importantly, it allows the flow network to be trained using only mono-modal flow supervision, while still contributing to accurate cross-modal alignment. To enable such supervision, we adopt a geometry-aware data synthesis pipeline that generates dense flow labels from single-view images. Considering that the synthetic data inevitably contains noise, we further adopt an outlier-robust loss to adaptively filter unreliable supervision based on residual magnitudes. These innovations enable effective flow network training without real-world labels, facilitating the decoupled training scheme. Compared to the conventional appearance-based optimization, our decoupled optimization strategy reduces the endpoint error (EPE) by 15.43 on the $MS^2$ (RGB-T) dataset.

Furthermore, we propose a cross-modal consistency constraint to jointly optimize both the modality transfer and flow estimation networks. Specifically, we apply spatial transformations to cross-modal image pairs, and enforce consistency between flow predictions before and after transformations. This constraint encourages direct learning of cross-modal flow, and strengthens the mutual promotion of the two networks, improving the EPE on the $MS^2$ (RGB-T) dataset from 4.81 to 3.46.

By integrating the above insights, DCFlow supports effective training of modern flow estimation networks such as RAFT (Teed & Deng, 2020) and FlowFormer (Huang et al., 2022), offering a general and effective network-agnostic training framework for cross-modal flow estimation. For comprehensive evaluation, we repurpose public multi-modal datasets by projecting LiDAR points to obtain ground-truth flow, creating five diverse datasets covering RGB, near-infrared (NIR), and thermal modalities. As shown in Fig. 1(b), DCFlow significantly surpasses existing unsupervised and large-scale pretrained approaches. In summary, our main contributions are as follows:

- We propose DCFlow, a general and network-agnostic self-supervised training framework for cross-modal flow estimation. DCFlow achieves state-of-the-art performance among all unsupervised approaches.

- We introduce a decoupled optimization strategy that enables single-modal flow supervision to benefit cross-modal flow estimation, supported by a geometry-aware data synthesis pipeline and an outlier-robust loss to reliably provide such supervision from single-view images.

- We devise a cross-modal consistency constraint to facilitate effective joint optimization of the modality transfer and flow estimation networks, significantly enhancing flow estimation accuracy.

- We construct a comprehensive cross-modal flow benchmark by repurposing publicly available datasets, covering diverse modalities such as RGB, NIR, and thermal.

## 2 RELATED WORK

**Cross-modal image matching.** Cross-modal image matching aims to establish spatial correspondences between images from different modalities, and has a wide range of applications (Jiang et al., 2021b; Li et al., 2024; Liu et al., 2024; Jiang et al., 2024). Traditional approaches (Shen et al., 2014; Kim et al., 2015) design cross-modal invariant descriptors, but often suffer from high computational cost. Recently, unsupervised deep learning methods (Arar et al., 2020; Wang et al., 2022; Xu et al., 2022) have been proposed, typically consisting of a modality transfer network and a flow estimation network. Due to the absence of ground-truth correspondences, these methods rely on appearance-based supervision, which often leads to ambiguity in textureless or repetitive regions. Alternatively, approaches such as CrossRAFT (Zhou et al., 2022) and MINIMA (Ren et al., 2025) synthesize cross-modal data using multi-view RGB images with known ground-truth displacements to provide direct motion supervision. However, the synthetic-to-real domain gap limits their generalization to real-world scenarios. Besides, we note that recent work SSHNet (Yu et al., 2025) introduces a split optimization framework for cross-modal homography estimation, which is limited to modeling global transformations. In contrast, we tackle the more challenging problem of dense pixel-wise correspondence estimation across modalities, which has broader applicability in real-world scenarios.

**Flow estimation.** Starting from FlowNet (Dosovitskiy et al., 2015), various network architectures (Teed & Deng, 2020; Jiang et al., 2021a; Huang et al., 2022) have been proposed under supervised learning, with most state-of-the-art methods adopting an iterative prediction paradigm based on cost volumes. In the unsupervised setting, prior work (Yu et al., 2016) introduces brightness constancy and motion smoothness as fundamental constraints. Subsequent approaches further enhance these ideas through specifically designed regularization strategies (Liu et al., 2020; Luo et al., 2021; Yuan et al., 2024), or design domain adaption techniques to enhance performance in adverse weather (Zhou et al., 2023a;b). Other methods (Watson et al., 2020; Aleotti et al., 2021; Liang et al., 2023; 2025) attempt to synthesize training data from single-view images. However, most of the above approaches are designed for and evaluated on RGB image pairs, highlighting the urgent need for effective solutions tailored to real-world cross-modal scenarios.

## 3 METHOD

### 3.1 PRELIMINARIES

This work tackles the problem of cross-modal flow estimation in an unsupervised setting. Given a cross-modal image pair $\mathbf{I}_A$ and $\mathbf{I}_B$ from modalities A and B respectively, our goal is to train a network $\mathcal{N}(\cdot)$ to predict the dense flow $\mathbf{F}_{B2A}$ from $\mathbf{I}_B$ to $\mathbf{I}_A$. $\mathcal{N}(\cdot)$ is typically decomposed into two components, formulated as

$$\mathbf{F}_{B2A} = \mathcal{N}(\mathbf{I}_A, \mathbf{I}_B) = \mathcal{F}_\theta(\mathcal{T}_\phi(\mathbf{I}_A), \mathbf{I}_B), \tag{1}$$

where $\mathcal{T}_\phi(\cdot)$ is a modality transfer network with learnable parameters $\phi$, which transforms $\mathbf{I}_A$ into modality B, and $\mathcal{F}_\theta(\cdot)$ is a mono-modal flow estimation network with learnable parameters $\theta$. Existing unsupervised approaches generally rely on photometric losses between the warped source and the target image for training, expressed as

$$\underset{\phi,\theta}{\arg\min} \, \mathcal{L}_{ph}\left(\mathcal{W}(\mathbf{I}_{A,T}, \mathbf{F}_{B2A}), \mathbf{I}_B\right), \tag{2}$$

where $\mathbf{I}_{A,T} = \mathcal{T}_\phi(\mathbf{I}_A)$ is the modality transferred image, $\mathcal{W}(\cdot)$ denotes the warping operation, and $\mathcal{L}_{ph}$ is a photometric similarity metric such as $L_1$ distance or SSIM. However, such supervision

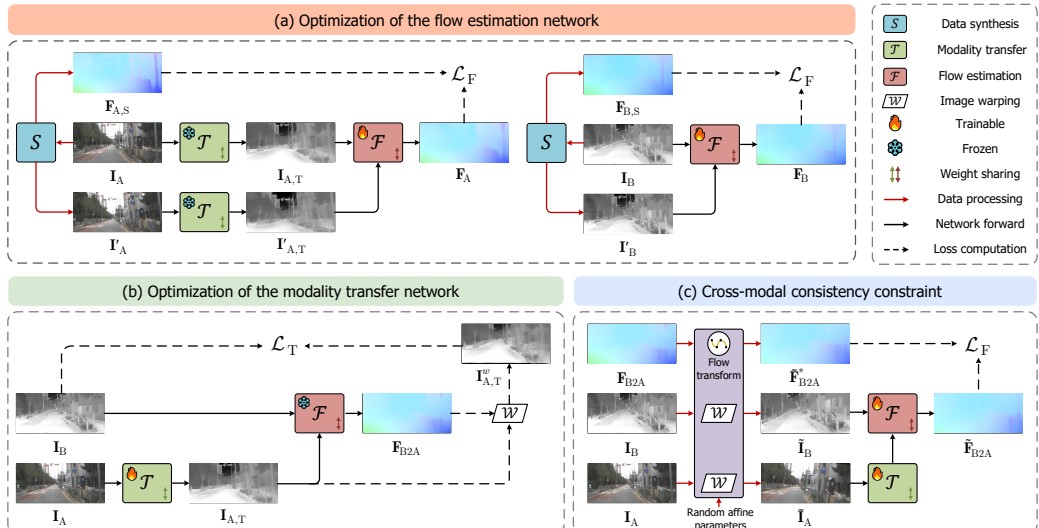

Figure 2: Schematic diagram of DCFlow, which incorporates a decoupled optimization strategy (a, b) and a cross-modal consistency constraint (c). (a) Optimization of the flow estimation network $\mathcal{F}_\theta$, where $\mathcal{F}_\theta$ is optimized using direct flow supervision from two-branch intra-modal synthetic data. (b) Optimization of the modality transfer network $\mathcal{T}_\phi$, where $\mathcal{T}_\phi$ learns to align modality $\mathbf{I}_A$ with $\mathbf{I}_B$ via perceptual similarity. (c) Cross-modal consistency constraint, where both networks are jointly optimized by enforcing flow consistency under known spatial transformations.

is inherently ambiguous in texture-less regions or repetitive patterns, and lacks direct motion cues, often leading to unsatisfactory results.

Motivated by these limitations, we believe that a potential improvement is to introduce motion supervision without using the cross-modal ground-truth flow. To achieve this goal, we propose DCFlow, a novel training framework that provides explicit flow supervision without requiring any labeled cross-modal data. We first decouple the overall task into modality transfer and mono-modal flow estimation, thus enabling mono-modal flow supervision to benefit cross-modal flow estimation. Furthermore, to ensure effective interaction between the two components, we introduce the cross-modal consistency constraint. This constraint facilitates collaboration between the networks, leading to improved performance. Fig. 2 illustrates the overall training framework of DCFlow.

### 3.2 DECOUPLED OPTIMIZATION

DCFlow adopts a decoupled optimization strategy for the modality transfer network $\mathcal{T}_\phi$ and the flow estimation network $\mathcal{F}_\theta$, which address the modality discrepancy and geometric misalignment between the input image pair separately, as illustrated in Fig. 2(a)(b). Each network is trained with task-specific supervision, forming a self-reinforcing process where improvements in one network provide better guidance for the other. Consequently, the training process becomes more stable, and converges to more accurate results.

**Flow estimation.** Fig. 2(a) illustrates the optimization process of the flow estimation network. In this stage, the weights of the modality transfer network $\mathcal{T}_\phi$ are frozen. The objective is to provide direct motion supervision for the flow network $\mathcal{F}_\theta$, addressing the limitations of implicit appearance-based supervision. Inspired by recent works (Zhang et al., 2024; Yu et al., 2025), we adopt a two-branch intra-modal supervision scheme using synthetic data from each modality. It enables the flow estimation network to process inputs from two different domains simultaneously at the start of training. Under the multi-task learning paradigm (Caruana, 1997), this setup progressively enhances the generalization of the network to cross-modal inputs, and facilitates convergence in a self-supervised manner.

Specifically, we construct two training triplets $(\mathbf{I}_A, \mathbf{I}'_A, \mathbf{F}_{A,S})$ and $(\mathbf{I}_B, \mathbf{I}'_B, \mathbf{F}_{B,S})$, where $\mathbf{I}'_{A/B}$ denotes the rendered novel view from $\mathbf{I}_{A/B}$, and $\mathbf{F}_{A/B,S}$ is the corresponding synthetic flow label. The

data synthesis pipeline is detailed in Sec. 3.3. The flow network is trained by minimizing the $L_1$ distance between the predicted and synthetic flows from both branches, formulated as

$$\underset{\theta}{\arg\min}(\mathcal{L}_{\mathrm{F}}(\mathbf{F}_{\mathrm{A}}, \mathbf{F}_{\mathrm{A,S}}) + \mathcal{L}_{\mathrm{F}}(\mathbf{F}_{\mathrm{B}}, \mathbf{F}_{\mathrm{B,S}})), \tag{3}$$

where the predicted flows are defined as $\mathbf{F}_{\mathrm{A}} = \mathcal{F}_{\theta}(\mathcal{T}_{\phi}(\mathbf{I}'_{\mathrm{A}}), \mathcal{T}_{\phi}(\mathbf{I}_{\mathrm{A}}))$ and $\mathbf{F}_{\mathrm{B}} = \mathcal{F}_{\theta}(\mathbf{I}'_{\mathrm{B}}, \mathbf{I}_{\mathrm{B}})$, and the loss $\mathcal{L}_{\mathrm{F}}$ is computed as

$$\mathcal{L}_{\mathrm{F}}(\mathbf{F}, \mathbf{F}_{\mathrm{S}}) = \|\mathbf{F} - \mathbf{F}_{\mathrm{S}}\|_1. \tag{4}$$

We implement $\mathcal{F}_{\theta}$ using RAFT (Teed & Deng, 2020), an iterative flow estimation architecture, and apply the loss over all intermediate predictions. Notably, DCFlow is agnostic to the choice of flow network, and supports alternative architectures, as demonstrated in Table 1e.

**Modality transfer.** Fig. 2(b) illustrates the optimization process of the modality transfer network. In this stage, the weights of the flow estimation network $\mathcal{F}_{\theta}$ are frozen. The modality transfer network $\mathcal{T}_{\phi}$ is optimized to translate the input image $\mathbf{I}_{\mathrm{A}}$ into the appearance of modality B.

Specifically, an estimated flow $\mathbf{F}_{\mathrm{B2A}}$ between the cross-modal image pair is used to warp the transferred image $\mathbf{I}_{\mathrm{A,T}}$, producing the warped output $\mathbf{I}^w_{\mathrm{A,T}} = \mathcal{W}(\mathbf{I}_{\mathrm{A,T}}, \mathbf{F}_{\mathrm{B2A}})$. The optimization objective is then given by

$$\underset{\phi}{\arg\min} \, \mathcal{L}_{\mathrm{T}}(\mathbf{I}^w_{\mathrm{A,T}}, \mathbf{I}_{\mathrm{B}}), \tag{5}$$

where $\mathcal{L}_{\mathrm{T}}$ is defined as the perceptual loss (Johnson et al., 2016), formulated as

$$\mathcal{L}_{\mathrm{T}}(\mathbf{I}^w_{\mathrm{A,T}}, \mathbf{I}_{\mathrm{B}}) = \sum_l \lambda_l \left\|\Phi_l(\mathbf{I}^w_{\mathrm{A,T}}) - \Phi_l(\mathbf{I}_{\mathrm{B}})\right\|_2, \tag{6}$$

with $\Phi_l(\cdot)$ denoting the $l$-th layer of a pretrained VGG network, and $\lambda_l$ the corresponding layer weight. The perceptual loss captures high-level structural and semantic similarity, and is less sensitive to spatial misalignments than pixel-wise metrics like $L_1$ distance. This makes it a robust supervisory signal for the modality transfer network, even when the estimated flow is imperfect, thereby guiding the training process toward a desired convergence. We implement $\mathcal{T}_{\phi}$ using a U-Net (Ronneberger et al., 2015) to preserve both fine-grained details and global contextual features.

Under this decoupled training strategy, the two networks are trained independently with stable supervision. Meanwhile, their outputs mutually reinforce each other, driving the entire framework toward continuous performance improvement. Better still, this strategy enables the flow network to be trained using only mono-modal supervision, while contributing to cross-modal alignment, fundamentally addressing limitations of appearance-based methods.

## 3.3 Data Synthesis and Outlier-robust Loss

To provide reliable supervision for the flow estimation network, we propose a geometry-aware data synthesis pipeline with an outlier-robust loss. The data synthesis process aims to generate a novel view image $\mathbf{I}'$ and its corresponding synthetic flow $\mathbf{F}_{\mathrm{S}}$ from a single input image $\mathbf{I}$. To achieve this, we introduce a lifting and reprojection technique that projects 2D pixels into 3D space and reprojects them into a virtual camera view. It produces photorealistic and geometrically consistent image pairs under realistic motion patterns, providing dense, geometry-grounded supervision without requiring multi-view images with ground-truth flow label. Fig. 3 illustrates the overall pipeline.

We first estimate a depth map $\mathbf{D}$ using a pretrained monocular depth model, such as UniDepth (Piccinelli et al., 2024). Each 2D pixel $\mathbf{x} = [u, v]^\top$ is projected into a 3D point $\mathbf{X} = [x, y, z]^\top$ via a sampled intrinsic matrix $\mathbf{K}$ as

$$\mathbf{X} \sim \mathbf{D}(\mathbf{x}) \cdot \mathbf{K}^{-1} \cdot \mathbf{x}. \tag{7}$$

For simplicity, we omit the homogeneous coordinate form. We then sample a virtual camera pose $\mathbf{T} \in \mathrm{SE}(3)$, and re-project the 3D points into the novel view as

$$\mathbf{x}' \sim \mathbf{K} \cdot \mathbf{T} \cdot \mathbf{X}, \tag{8}$$

where $\mathbf{x}' = [u', v']^\top$. Then, the rendered image $\mathbf{I}'$ is obtained by sampling the corresponding pixel values from $\mathbf{I}$, and the synthetic flow $\mathbf{F}_{\mathrm{S}}$ is defined as the 2D displacement between $\mathbf{x}$ and $\mathbf{x}'$.

Figure 3: Illustration of the geometry-aware data synthesis pipeline.

To identify occluded and invisible regions introduced by viewpoint changes, we adopt a photometric consistency check to generate a binary valid mask $\mathbf{M}$, where $\mathbf{M} = 1$ indicates valid pixels. Specifically, we warp the rendered image $\mathbf{I}'$ back to the original view using $\mathbf{F}_S$, and measure the photometric error against $\mathbf{I}$. Pixels with photometric errors exceeding a threshold are marked as invalid and excluded from loss computation. The flow loss is then defined as

$$\mathcal{L}_F = \frac{1}{\sum \mathbf{M}} \sum_{\mathbf{M}(\mathbf{x})=1} \|\mathbf{F}(\mathbf{x}) - \mathbf{F}_S(\mathbf{x})\|_1. \tag{9}$$

While the valid mask effectively excludes occluded and invisible regions, some pixels may still yield unreliable supervision due to rendering artifacts or depth estimation errors. These regions, such as distorted boundaries or thin structures, may contain valid flow but poor appearance quality, making them unsuitable for flow supervision.

To address this, in line with previous approaches (Wang et al., 2021; Keetha et al., 2025; Ding et al., 2025), we adopt an outlier-robust loss that further filters noisy supervision by discarding a small fraction of pixels with the highest residuals. Specifically, we sort the per-pixel $L_1$ flow error within valid regions, and remove the top-$\tau\%$ pixels for loss computation. This ensures that the supervision focuses on regions with both accurate motion and relatively high appearance fidelity. The outlier-robust loss is defined as

$$\mathcal{L}_F = \frac{1}{|\Omega_\tau|} \sum_{\mathbf{x} \in \Omega_\tau} \|\mathbf{F}(\mathbf{x}) - \mathbf{F}_S(\mathbf{x})\|_1, \tag{10}$$

where $\Omega_\tau$ denotes the set of valid pixels after excluding the top-$\tau\%$ largest residuals. It provides a more robust training signal by avoiding noisy supervision caused by uncertain depth or texture distortions, thereby further improving the stability and accuracy of flow learning.

### 3.4 CROSS-MODAL CONSISTENCY CONSTRAINT

Though the decoupled optimization strategy ensures stable convergence, it does not explicitly teach the network to perform cross-modal flow estimation. Instead, the model relies on the generalization ability of two independently optimized networks. To overcome this limitation, we introduce a cross-modal consistency constraint that jointly optimizes both networks by enforcing flow consistency under known spatial transformations, as illustrated in Fig. 2(c).

Given a cross-modal image pair $(\mathbf{I}_A, \mathbf{I}_B)$ and the corresponding flow prediction $\mathbf{F}_{B2A}$, we apply random affine transformations (*e.g.*, scaling, rotation, and translation) to both images, yielding the augmented image pair $(\tilde{\mathbf{I}}_A, \tilde{\mathbf{I}}_B)$. Since the transformations are known, we can derive a transformed flow $\tilde{\mathbf{F}}^*_{B2A}$ from $\mathbf{F}_{B2A}$. The augmented pair is then passed through the full network pipeline, producing a new prediction $\tilde{\mathbf{F}}_{B2A} = \mathcal{F}_\theta(\mathcal{T}_\phi(\tilde{\mathbf{I}}_A), \tilde{\mathbf{I}}_B)$. Based on the assumption that flow predictions should remain geometrically consistent under known transformations, the joint optimization objective is defined as

$$\underset{\phi,\theta}{\arg\min} \, \mathcal{L}_F(\tilde{\mathbf{F}}_{B2A}, \tilde{\mathbf{F}}^*_{B2A}), \tag{11}$$

where $\mathcal{L}_F$ is the outlier-robust loss in Eq. 10. This self-supervised constraint enables direct learning of cross-modal flow estimation, encouraging mutual adaptation between the two networks and thus improving final performance.

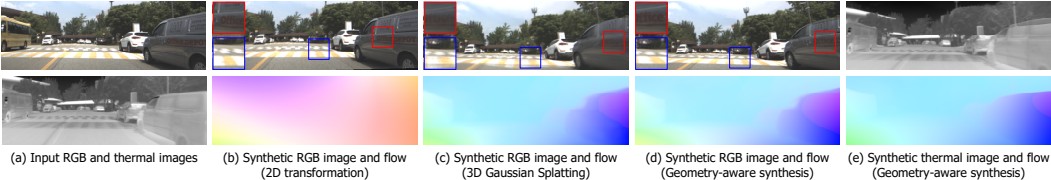

(a) Input RGB and thermal images    (b) Synthetic RGB image and flow (2D transformation)    (c) Synthetic RGB image and flow (3D Gaussian Splatting)    (d) Synthetic RGB image and flow (Geometry-aware synthesis)    (e) Synthetic thermal image and flow (Geometry-aware synthesis)

Figure 4: Qualitative comparison of different synthetic flow data generation strategies.

### 3.5 OVERALL TRAINING OBJECTIVE

The entire training objective can be formulated as

$$\underset{\phi,\theta}{\arg\min} \, \mathcal{L}_{F}(\mathbf{F}_{A}, \mathbf{F}_{A,S}) + \mathcal{L}_{F}(\mathbf{F}_{B}, \mathbf{F}_{B,S}) + \lambda_{T}\mathcal{L}_{T}(\mathbf{I}_{A,T}^{w}, \mathbf{I}_{B}) + \lambda_{C}\mathcal{L}_{F}(\tilde{\mathbf{F}}_{B2A}, \tilde{\mathbf{F}}_{B2A}^{*}), \quad (12)$$

where $\lambda_{T}$ and $\lambda_{C}$ denote the loss weights for the modality transfer and cross-modal consistency components, respectively. We note that these three losses are jointly optimized within a single gradient descent step, with each loss imposed on its corresponding set of parameters. Once combined, the decoupled optimization allows each network to learn its specific task robustly, while the consistency constraint enhances collaboration between two networks for better cross-modal flow estimation, ultimately leading to stable and effective self-supervised training.

For implementation, we set $\lambda_{T} = 2.0$ and $\lambda_{C} = 0.05$. We train the entire network from scratch for $30,000$ iterations with a batch size of $4$. The cross-modal consistency constraint is introduced after $10,000$ iterations, allowing the model to produce reliable flow predictions before joint optimization begins.

## 4 EXPERIMENTS

### 4.1 EXPERIMENTAL SETTING

**Datasets.** We evaluate DCFlow on three public datasets, *i.e.*, MS$^2$ (Shin et al., 2023), VTD (Guo et al., 2023), and RNS (Kim & Baek, 2025). These datasets provide multi-modal data including RGB, near-infrared (NIR), thermal (T), and LiDAR, and consist of multiple video sequences. To adapt them for cross-modal flow evaluation, we repurpose the raw data using the following procedure. We first resize and crop images across modalities to achieve consistent effective focal lengths, based on the provided intrinsic parameters. Then, we project LiDAR points from one image to another using the known extrinsic parameters, and compute the ground-truth flow from the 2D displacements of valid projected points. Due to the inherent sparsity of LiDAR, the resulting flow labels are sparse. For dataset splitting, we use the first 80% frames in each video sequence for training and the remaining 20% for testing. Notably, the MS$^2$ dataset simultaneously captures RGB, NIR, and T modalities, allowing us to construct three sub-datasets for comprehensive evaluation under different modality gaps. We denote these as MS$^2$ (RGB-T), MS$^2$ (RGB-NIR), and MS$^2$ (NIR-T).

**Metrics.** We report the endpoint error (EPE) and the flow outlier rate (F1). EPE measures the average $L_2$ distance between the predicted flow and the ground-truth, while F1 denotes the percentage of pixels with EPE greater than both 3 pixels and 5% of the ground-truth magnitude. Lower EPE and F1 values indicate better performance.

### 4.2 ABLATION STUDY

Ablation studies are conducted on the MS$^2$ (RGB-T) dataset unless otherwise specifically stated.

**Training strategy.** Table 1a presents the ablation study on different training strategies of DCFlow. We start with the appearance-based optimization baseline, where the entire network is trained using only the photometric similarity loss in Eq. 2. As shown, this strategy leads to poor performance, reflecting the inherent limitations of relying solely on appearance cues. We then evaluate our decoupled optimization strategy, which separately trains the modality transfer and flow estimation

Table 1: Ablation studies of DCFlow on the MS$^2$ (RGB-T) dataset.

(a) Ablation study on training strategies.

| Training strategy | EPE | F1 |
|---|---|---|
| Appearance-based optimization | 21.23 | 98.45 |
| Decoupled optimization | 5.80 | 57.18 |
| + Outlier-robust loss | 4.81 | 51.39 |
| + Cross-modal consistency constraint | **3.46** | **35.89** |

(b) Ablation study on data synthesis strategies.

| Data synthesis strategy | EPE | F1 |
|---|---|---|
| 2D transformation | 13.12 | 95.21 |
| 3D Gaussian Splatting | 5.11 | 63.12 |
| Geometry-aware synthesis (Ours) | **3.46** | **35.89** |

(c) Ablation study on inaccurate depth estimation.

| Depth quality | $\sigma$ of Gaussian noise | $\sigma$ of scaling noise | Kernel size of blurring | EPE | F1 |
|---|---|---|---|---|---|
| Network output | - | - | - | **3.46** | **35.89** |
| Small degradation | 0.1 | 0.1 | 7 | 3.81 | 40.66 |
| Large degradation | 0.2 | 0.2 | 14 | 4.00 | 49.08 |

(d) Ablation study on different depth estimation networks.

| Depth network | EPE | F1 |
|---|---|---|
| Metric3D v2 | 3.74 | 39.41 |
| Depth Anything v2 | **3.45** | **34.81** |
| UniDepth (Default) | 3.46 | 35.89 |

(e) Ablation study on different flow networks.

| Training strategy | RAFT | | GMA | | FlowFormer | | SEA-RAFT | |
|---|---|---|---|---|---|---|---|---|
| | EPE | F1 | EPE | F1 | EPE | F1 | EPE | F1 |
| Appearance-based optimization | 21.23 | 98.45 | 24.27 | 97.39 | 29.22 | 99.43 | 23.93 | 98.66 |
| DCFlow (Ours) | **3.46** | **35.89** | **4.13** | **48.04** | **3.66** | **37.97** | **3.57** | **38.63** |

Table 2: Ablation study on the direction of modality transfer.

| Dataset | MS$^2$ (RGB-T) | | MS$^2$ (NIR-T) | | MS$^2$ (RGB-NIR) | |
|---|---|---|---|---|---|---|
| Direction | RGB→T | T→RGB | NIR→T | T→NIR | RGB→NIR | NIR→RGB |
| EPE | 3.46 | 3.68 | 4.53 | 4.80 | 0.96 | 0.94 |
| F1 | 35.89 | 44.11 | 49.81 | 60.71 | 5.00 | 4.66 |

networks using task-specific objectives. In this setup, synthetic flow data is introduced to enable explicit motion supervision for the flow network. This strategy yields stable convergence and achieves an EPE of 5.80. Introducing the outlier-robust loss further improves performance, reducing EPE by 0.99 and F1 by 5.79, demonstrating its effectiveness in suppressing noisy supervision from synthetic artifacts. Finally, incorporating the proposed cross-modal consistency constraint leads to the best performance, with an EPE of 3.46 and F1 of 35.89. These results confirm that enforcing spatial consistency during joint optimization significantly enhances flow estimation accuracy.

**Flow data synthesis.** DCFlow introduces a geometry-aware synthesis pipeline to generate image pairs with dense flow labels from a single image. As alternatives, we evaluate the 2D transformation (*e.g.*, homography) and feed-forward 3D Gaussian Splatting (Szymanowicz et al., 2025) as substitute data generation strategies. We present qualitative comparisons of these strategies in Fig. 4. The 2D transformation (Fig. 4(b)) ignores scene geometry and produces unrealistic motion patterns, which cause the model to overfit to such distortions. The 3D Gaussian Splatting approach (Fig. 4(c)) synthesizes novel views with 3D awareness, but often suffers from visual artifacts and instability caused by imperfect Gaussian primitives estimation. In contrast, our proposed data synthesis pipeline (Fig. 4(d)) produces geometrically consistent and visually plausible novel views, offering more reliable training data for flow networks. Moreover, Fig. 4(e) shows that our geometry-aware synthesis pipeline generalizes well to challenging modalities such as thermal images. Table 1b reports the results trained under each data synthesis pipeline, further demonstrating that our geometry-aware synthesis significantly outperforms the alternatives.

**Discussion on monocular depth estimation.** In DCFlow, we adopt a pretrained monocular depth model to estimate the depth of input images for the geometry-aware data synthesis pipeline. We note that DCFlow is robust to imperfect or biased depth estimation. In general, the estimated depth serves only as an intermediate variable for flow data synthesis. Although the depth prediction may not be perfect, it is sufficient to represent the relative distance relationships between different regions of an image. Since both the warped image and the corresponding synthetic flow are computed using the same depth prediction, the synthetic flow label can still accurately represent the motion between the source image and the warped image. In addition, the photometric consistency checking and the outlier-robust loss filter out invisible areas and regions with artifacts, which ensures that the supervision focuses on pixels with reliable motion and appearance. Moreover, the cross-modal consistency constraint provides an additional source of flow supervision that does not rely on monocular depth. To further demonstrate the robustness of DCFlow under inaccurate depth, we add degradations to the depth map estimated by UniDepth (Piccinelli et al., 2024) for flow synthesis, including Gaussian noise, scaling noise, and edge blurring. As shown in Table 1c, DCFlow shows only slight performance drop even when the depth maps are heavily corrupted, and it still performs favorably compared with existing unsupervised baselines.

Besides, we note that the monocular depth for NIR and thermal images is obtained by directly applying depth foundation models pretrained on the RGB domain in a zero-shot manner. Although these depth models are trained on RGB data, prior works (Shin et al., 2023; Shin & Park, 2025) show that monocular depth estimation mainly relies on geometry cues such as perspective geometry, occlusion boundaries, and texture gradients, which are largely modality invariant. Moreover, these depth foundation models are trained on diverse datasets with strong augmentations such as brightness, contrast, and hue changes, which further enhance their generalization ability to unseen modalities. As a result, models like UniDepth generalize reasonably well to NIR and thermal inputs. We replace UniDepth (Piccinelli et al., 2024) with Depth Anything v2 (Yang et al., 2024) and Metric3D v2 (Hu et al., 2024) for flow synthesis, and the results in Table 1d show that DCFlow consistently achieves strong performance across different depth networks.

**Generalization ability for different flow networks.** We replace RAFT (Teed & Deng, 2020) with GMA (Jiang et al., 2021a), FlowFormer (Huang et al., 2022), and SEA-RAFT (Wang et al., 2024), and report the results in Table 1e. Our DCFlow consistently achieves superior performance across different flow networks, demonstrating strong generalization and compatibility.

**Direction of modality transfer.** Since some modalities contain more information, transferring from richer to less informative modalities could make the learning process easier. We select the transfer direction based on the observation that modalities such as RGB and NIR contain richer texture and structure than thermal, making the modality transfer easier and more stable than the reverse direction. As shown in Table 2, transferring from RGB/NIR to thermal outperforms the reverse direction. The performance difference between RGB and NIR mapping is relatively small.

## 4.3 Comparisons with Existing Approaches

**Baselines.** We evaluate our DCFlow with large-scale pretrained approaches including Cross-RAFT (Zhou et al., 2022) and MINIMA (Ren et al., 2025), unsupervised approaches including NeMAR (Arar et al., 2020) and UMF-CMGR (Wang et al., 2022), and supervised approaches including RAFT (Teed & Deng, 2020), GMA (Jiang et al., 2021a), FlowFormer (Huang et al., 2022), and SEA-RAFT (Wang et al., 2024). CrossRAFT and MINIMA are pretrained on large-scale synthetic datasets with ground-truth flow label. We evaluate their performance using publicly available checkpoints. We retrain all unsupervised and supervised baselines under the same settings as ours for fairness. The supervised approaches are trained using sparse ground-truth flow annotations.

**Quantitative comparison.** Table 3 reports the quantitative results on five cross-modal datasets. Among all unsupervised and large-scale pretrained approaches, our DCFlow consistently achieves the best performance on both metrics, demonstrating strong generalization ability across diverse modalities. The existing unsupervised approaches such as NeMAR and UMF-CMGR generally yield unsatisfactory results, which aligns with findings from prior studies (Zhang et al., 2024; Yu et al., 2025). This suggests that appearance-based optimization struggles to converge under significant modality discrepancy and geometric misalignment. In contrast, DCFlow produces stable and accurate flow estimation, highlighting the effectiveness of our training framework. Compared to

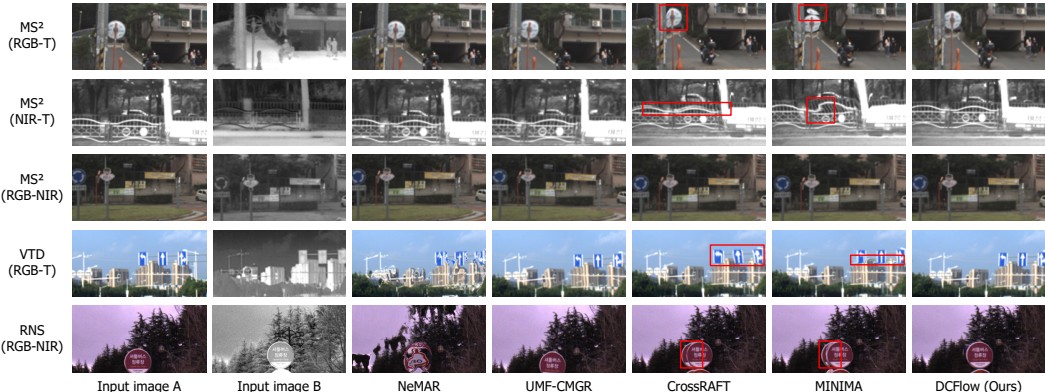

Figure 5: Qualitative comparison of DCFlow and other approaches. The first two columns show the input image pairs, and the remaining columns visualize the image from modality A warped using the estimated flow from each approach. For clarity, cropped patches of the full-resolution images are shown. The red boxes highlight the distortion regions.

Table 3: Quantitative comparison of DCFlow and other approaches. The EPE and F1 are reported. The best results among all large-scale pretrained and unsupervised methods are highlighted in bold.

| Category | Method | $MS^2$ (RGB-T) | | $MS^2$ (NIR-T) | | $MS^2$ (RGB-NIR) | | VTD (RGB-T) | | RNS (RGB-NIR) | |
|---|---|---|---|---|---|---|---|---|---|---|---|
| | | EPE | F1 | EPE | F1 | EPE | F1 | EPE | F1 | EPE | F1 |
| Supervised | RAFT | 1.70 | 14.76 | 1.80 | 16.50 | 0.23 | 0.57 | 1.14 | 8.45 | 1.63 | 9.32 |
| | GMA | 1.67 | 14.58 | 1.80 | 16.55 | 0.24 | 0.61 | 1.15 | 8.67 | 1.53 | 6.53 |
| | FlowFormer | 1.65 | 14.28 | 1.78 | 16.22 | 0.25 | 0.77 | 1.20 | 8.43 | 1.87 | 13.88 |
| | SEA-RAFT | 1.65 | 14.04 | 1.97 | 17.64 | 0.21 | 0.48 | 1.31 | 9.49 | 1.37 | 6.42 |
| Large-scale pretrained | CrossRAFT | 6.21 | 70.20 | 7.06 | 73.06 | 4.32 | 29.54 | 9.86 | 89.33 | 2.04 | 15.53 |
| | MINIMA | 5.97 | 66.12 | 7.10 | 70.37 | 5.44 | 33.48 | 6.34 | 80.41 | 2.34 | 15.18 |
| Unsupervised | NeMAR | 19.25 | 99.80 | 28.41 | 99.99 | 11.39 | 99.92 | 23.43 | 96.89 | 25.11 | 99.83 |
| | UMF-CMGR | 18.84 | 99.78 | 26.67 | 99.99 | 8.85 | 99.27 | 28.05 | 99.13 | 31.13 | 99.98 |
| | DCFlow (Ours) | **3.46** | **35.89** | **4.53** | **49.81** | **0.96** | **5.00** | **3.65** | **48.49** | **1.90** | **13.05** |

large-scale pretrained approaches like CrossRAFT and MINIMA, DCFlow achieves significantly better results. For instance, DCFlow yields 42.0%, 36.2%, 82.4%, 42.4%, and 18.8% lower EPEs than MINIMA on the five datasets, demonstrating the advantage of learning directly from unlabeled real-world data over relying on synthetic cross-modal datasets. When compared with supervised approaches, our DCFlow achieves competitive performance, despite the absence of using ground-truth cross-modal flow labels. These results highlight the effectiveness of DCFlow.

**Qualitative comparison.** Fig. 5 presents qualitative results on five datasets. We visualize the image from modality A warped using the estimated flow to assess the accuracy of each method. As shown, previous unsupervised approaches such as NeMAR and UMF-CMGR struggle with large cross-modal misalignments. Although CrossRAFT and MINIMA produce coarse alignment, they still exhibit noticeable mismatches in several regions, as highlighted by the red boxes. In contrast, DCFlow achieves more precise warping, demonstrating the superiority of our training framework.

## 5 CONCLUSIONS

We have presented DCFlow, a novel self-supervised framework for cross-modal flow estimation that combines a decoupled optimization strategy and a cross-modal consistency constraint. The former tackles modality discrepancy and geometric misalignment with task-specific supervision, while the latter enables direct learning of cross-modal flow. Within this framework, we introduce a geometry-aware synthesis pipeline with an outlier-robust loss to provide reliable flow supervision from single-view images. Experiments on multiple datasets demonstrate that DCFlow achieves state-of-the-art performance among unsupervised approaches.

ACKNOWLEDGMENTS

This work was supported in part by the National Natural Science Foundation of China under grants 62301484 and 62571478, in part by the Ningbo Natural Science Foundation of China under grant 2024J454, in part by the Young Talent Fund of Zhejiang Association for Science and Technology under grant ZJSKXQT2026135, and in part by the Jinhua Science and Technology Bureau Project under grant 2026-1-022.

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

# A Appendix

## A.1 Network Architectures

In this paper, we propose DCFlow, a novel self-supervised framework for cross-modal flow estimation. It serves as a general and network-agnostic training paradigm, supporting the effective training of modern flow estimation networks. In our main experiments, we adopt U-Net (Ronneberger et al., 2015) for modality transfer and RAFT (Teed & Deng, 2020) for flow estimation.

**Modality transfer.** As illustrated in Fig. 6(a), the modality transfer network adopts an encoder-decoder structure with skip connections to preserve spatial details, and operates at four resolution scales. We use two convolutional layers with batch normalization and ReLU activations as the basic unit, max-pooling for downsampling, and bilinear interpolation for upsampling. Given the input image $\mathbf{I} \in \mathbb{R}^{H \times W \times 3}$, the feature maps across the four scales have sizes of $H \times W \times C$, $\frac{H}{2} \times \frac{W}{2} \times 2C$, $\frac{H}{4} \times \frac{W}{4} \times 4C$, and $\frac{H}{8} \times \frac{W}{8} \times 8C$, respectively. The U-Net architecture allows the network to translate images from one modality to another without any task-specific modifications, demonstrating the generalization ability and robustness of our training framework.

**Flow estimation.** We adopt RAFT as the flow estimation network. As shown in Fig. 6(b), it consists of a feature encoder, a context encoder, a correlation layer, and an iterative flow decoder. Given the input image pair $\mathbf{I}_1, \mathbf{I}_2 \in \mathbb{R}^{H \times W \times 3}$, the goal is to predict the optical flow $\mathbf{F} \in \mathbb{R}^{H \times W \times 2}$. The feature encoder extracts matching features from both images, while the context encoder processes only $\mathbf{I}_1$ to extract contextual information. The correlation layer then computes a 4D cost volume by taking the inner product between all pairs of matching features. Finally, the cost volume and context features are passed into the iterative decoder, which progressively refines the flow predictions. For more details, please refer to RAFT (Teed & Deng, 2020).

## A.2 Details of Cross-modal Consistency Constraint

In the following, we describe the details on how the transformed flow is obtained. Given a cross-modal image pair $(\mathbf{I}_A, \mathbf{I}_B)$ and the predicted flow $\mathbf{F}_{B2A}$, we apply the same random affine transformation $\mathcal{A}(\mathbf{x}) = \mathbf{A}\mathbf{x} + \mathbf{t}$ to both images, where $\mathbf{A}$ encodes rotation and scaling, and $\mathbf{t}$ is the translation. We denote the original flow as $\mathbf{F}_{B2A}(\mathbf{x}_B) = \mathbf{x}_A - \mathbf{x}_B$. After transformation, the corresponding points are given by $\tilde{\mathbf{x}}_B = \mathcal{A}(\mathbf{x}_B)$ and $\tilde{\mathbf{x}}_A = \mathcal{A}(\mathbf{x}_A)$. The transformed flow $\tilde{\mathbf{F}}_{B2A}^*$ can be formulated as

$$\tilde{\mathbf{F}}_{B2A}^*(\tilde{\mathbf{x}}_B) = \tilde{\mathbf{x}}_A - \tilde{\mathbf{x}}_B = \mathbf{A}(\mathbf{x}_A - \mathbf{x}_B) = \mathbf{A}\mathbf{F}_{B2A}(\mathbf{x}_B). \tag{13}$$

We note that only the linear part $\mathbf{A}$ affects the flow, while the translation term $\mathbf{t}$ cancels out. In practice, for each $\tilde{\mathbf{x}}_B$, we sample the corresponding $\mathbf{F}_{B2A}(\mathbf{x}_B)$ via bilinear interpolation and then multiply it by $\mathbf{A}$ to obtain the transformed flow.

## A.3 More Implementation Details

We implement DCFlow using PyTorch. The channel dimension $C$ in the modality transfer network is set to 16. The number of iterations in RAFT is fixed to 6. The threshold of the photometric error in the data synthesis pipeline is set to 10, defined on 8-bit image intensities. The top-$\tau\%$ threshold in the outlier-robust loss for flow estimation is set to 20%. For the cross-modal consistency constraint, we apply random affine transformations, including rotations within $\pm 3°$, scaling factors in the range of $[0.95, 1.05]$, and translations within $\pm 24$ pixels along both axes. We adopt the AdamW optimizer with a maximum learning rate of 0.0004, and apply the cosine decay schedule during training. All experiments are conducted on NVIDIA RTX 4090 GPUs.

## A.4 Metrics

We use the endpoint error (EPE) and flow outlier rate (F1) as quantitative metrics to evaluate flow accuracy.

(a) Modality transfer network

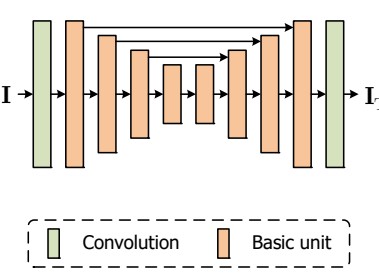

(b) Flow estimation network

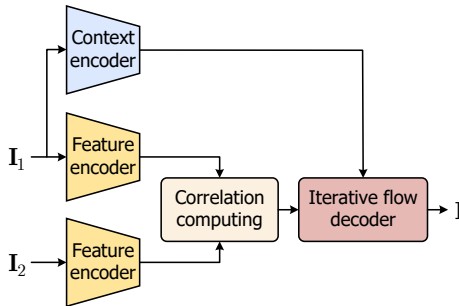

Figure 6: Illustration of the network architectures. (a) Modality transfer network, which adopts a U-Net architecture. (b) Flow estimation network, which adopts RAFT.

Table 4: The statistics of cross-modal flow datasets.

| Dataset | Modality | # Training | # Testing | Resolution |
|---------|----------|-----------|-----------|------------|
| MS$^2$ | RGB-T-NIR | 3631 | 907 | 608×192 |
| VTD | RGB-T | 1536 | 392 | 512×320 |
| RNS | RGB-NIR | 1187 | 294 | 704×512 |

The EPE is defined as the average $L_2$ distance between the predicted optical flow $\mathbf{F}$ and the ground-truth flow $\mathbf{F}_{GT}$ over all valid pixels, formulated as

$$\text{EPE} = \frac{1}{N} \sum_{\mathbf{x}} \|\mathbf{F}(\mathbf{x}) - \mathbf{F}_{GT}(\mathbf{x})\|_2, \qquad (14)$$

where $\mathbf{x}$ indexes valid pixel locations, and $N$ is the total number of valid pixels. Lower EPE indicates higher accuracy.

The F1 score measures the percentage of outlier pixels where the EPE exceeds both 3 pixels and 5% of the ground-truth flow magnitude. A pixel $\mathbf{x}$ is considered an outlier if

$$\|\mathbf{F}(\mathbf{x}) - \mathbf{F}_{GT}(\mathbf{x})\|_2 > \max(3,\ 0.05 \cdot \|\mathbf{F}_{GT}(\mathbf{x})\|_2). \qquad (15)$$

The F1 score is then computed as

$$\text{F1} = \frac{1}{N} \sum_{\mathbf{x}} \mathbf{1}\left[\mathbf{x} \text{ is an outlier}\right] \times 100\%, \qquad (16)$$

where $\mathbf{1}[\cdot]$ denotes the indicator function. Lower F1 indicates better robustness to large flow errors.

### A.5 Details of Datasets

We evaluate DCFlow on MS$^2$ (Shin et al., 2023), VTD (Guo et al., 2023), and RNS (Kim & Baek, 2025) datasets. Specifically, the MS$^2$ dataset contains three modality pairs, namely MS$^2$ (RGB-T), MS$^2$ (RGB-NIR), and MS$^2$ (NIR-T). Dataset statistics are summarized in Table 4. In the following, we describe the process of dataset repurposing.

To ensure scale consistency across modalities, we first standardize the image resolution, and unify the effective camera intrinsics. This ensures that all modalities share a common focal length scale, enabling accurate reprojection between viewpoints. Formally, we adjust the intrinsic matrices $\mathbf{K}_A$ and $\mathbf{K}_B$ of modalities A and B, respectively, to a shared intrinsic matrix $\hat{\mathbf{K}}$:

$$\mathbf{K}_A \to \hat{\mathbf{K}}, \quad \mathbf{K}_B \to \hat{\mathbf{K}}. \qquad (17)$$

After aligning the intrinsics, we compute ground-truth optical flow between modality pairs using LiDAR depth and known extrinsic calibration. For each valid pixel $\mathbf{x}_B = [u, v]^\top$ in modality B with depth $z$, we first back-project it into 3D space, expressed as

$$\mathbf{X}_B = z \cdot \hat{\mathbf{K}}^{-1} \begin{bmatrix} u \\ v \\ 1 \end{bmatrix}. \qquad (18)$$

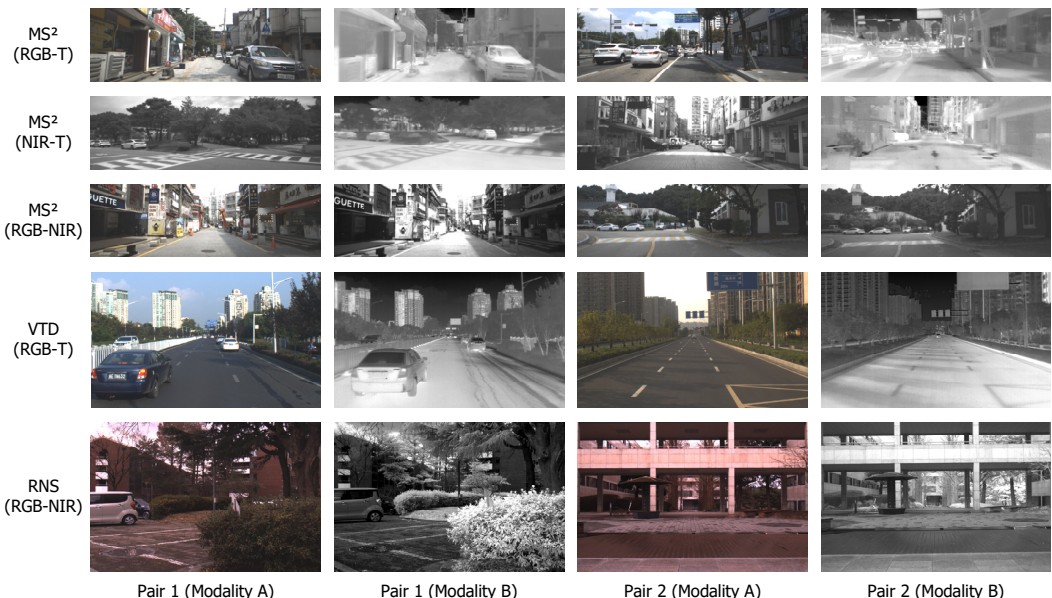

Figure 7: Cross-modal input image pairs from each dataset. For each dataset, we present two examples.

Table 5: Ablation on the value of $\tau$ in the outlier-robust loss.

| $\tau$ | EPE | F1 |
|---|---|---|
| w/o outlier-robust loss | 5.80 | 57.18 |
| 10 | 5.40 | 55.59 |
| 20 (Default) | **4.81** | 51.39 |
| 30 | 5.01 | **50.22** |
| 40 | 5.59 | 58.29 |

We then transform the 3D point into the coordinate system of modality A using the extrinsic matrix $\mathbf{T}_{B \to A} = [\mathbf{R}, \mathbf{t}] \in SE(3)$, formulated as

$$\mathbf{X}_A = \mathbf{R} \cdot \mathbf{X}_B + \mathbf{t}. \tag{19}$$

The corresponding 2D projection in modality A is given by

$$\mathbf{x}_A = \pi(\hat{\mathbf{K}} \cdot \mathbf{X}_A), \tag{20}$$

where $\pi(\cdot)$ denotes the perspective projection. The ground-truth flow is then computed as the 2D displacement between corresponding points as

$$\mathbf{F}_{B2A}(\mathbf{x}_B) = \mathbf{x}_A - \mathbf{x}_B. \tag{21}$$

Only pixels with valid depth and successful projections within image boundaries are retained. We present two cross-modal image pairs for each dataset in Fig. 7.

## A.6 ADDITIONAL ABLATION STUDIES

**Ablation on the value of $\tau$ in the outlier-robust loss.** We vary $\tau$ from 10 to 40, and list the results in Table 5. When $\tau$ is too small, the filtering of undesired regions becomes insufficient, and noisy pixels still contribute to supervision. When $\tau$ is too large, too many pixels are removed, and the supervision signal becomes weak. We choose $\tau$ as 20 to provide a good balance between removing unreliable regions and keeping enough valid information for supervision.

**Comparison with other robust loss functions.** We compare our outlier robust loss with the Charbonnier loss and the Mixture of Laplace loss used in SEA-RAFT (Wang et al., 2024). As shown

Table 6: Comparison with other robust loss functions.

| Loss type | EPE | F1 |
|---|---|---|
| L1 loss (baseline) | 5.80 | 57.18 |
| Charbonnier loss | 6.02 | 63.62 |
| Mixture-of-Laplace loss | 5.94 | 56.21 |
| Outlier-robust loss (Ours) | **4.81** | **51.39** |

Table 7: Ablation on the affine ranges in the cross-modal consistency constraint.

| Affine range | Rotation | Translation | Scaling | EPE | F1 |
|---|---|---|---|---|---|
| w/o consistency constraint | - | - | - | 4.81 | 51.39 |
| Small | [-1, +1] | [-12, +12] | [0.98, 1.02] | 3.61 | 36.08 |
| Medium (Default) | [-3, +3] | [-24, +24] | [0.95, 1.05] | **3.46** | **35.89** |
| Large | [-5, +5] | [-36, +36] | [0.92, 1.08] | 3.93 | 46.37 |

in Table 6, both alternatives lead to inferior results, while our outlier-robust loss achieves the best performance by more effectively suppressing high error outliers.

**Ablation on the affine ranges in the cross-modal consistency constraint.** We evaluate small, medium, and large affine perturbations for consistency constraint, and list the results in Table 7. It can be seen that small and medium perturbations already introduce sufficient geometric variation for the consistency constraint, while excessively large perturbations may create unrealistic deformations that harm the supervision quality. We choose medium perturbations as default.

## A.7 MORE EXPERIMENTAL RESULTS

**Generalization to unseen domains.** We present the cross-dataset generalization results between $MS^2$(RGB–T) and VTD (RGB–T) in Table 8. When trained on one dataset and evaluated on another unseen dataset, DCFlow achieves better performance than existing unsupervised approaches trained directly on the target dataset. This highlights the advantage of our framework over previous unsupervised approaches that rely solely on appearance-based supervision. Moreover, although large-scale pretrained approaches use substantially larger synthetic datasets with ground-truth flow, DCFlow achieves comparable or better generalization under the cross-dataset evaluation setting, while requiring far less training data.

**Generalization to unseen sensor types.** We evaluate the cross-modal generalization ability of DCFlow by testing models trained on $MS^2$(RGB–T) and $MS^2$(RGB–NIR) on the unseen $MS^2$(NIR–T) modality. As shown in Table 9, the model trained on RGB–T achieves reasonable performance on NIR–T, whereas the model trained on RGB–NIR fails to generalize. This can likely be attributed to the modality gap between RGB–T and NIR–T is relatively closer, making the transfer from RGB–T to NIR–T more feasible. In contrast, the gap between RGB–NIR and NIR–T is substantially larger, which leads to poor generalization.

**Modality transfer results.** We present qualitative results of modality transfer on each dataset in Fig. 8. The translated images preserve the structural details of the source modality, while showing appearance characteristics aligned with the target modality. These results clearly demonstrate the effectiveness of our proposed learning framework.

**Visualization of synthetic flow data.** Fig. 9 presents examples of synthetic flow generated from single-view images. The results demonstrate that our geometry-aware data generation pipeline produces high-quality flow supervision, and generalizes well across different modalities.

**Visualization of the valid mask used for flow supervision.** We present examples of the valid mask used for flow supervision in Fig. 10. Though the imperfect depth estimation would lead to visual artifacts in the synthetic images, the photometric consistency checking and the outlier-robust loss effectively remove these regions for loss computation, enabling high-quality supervision of the flow network.

Table 8: Cross-dataset generalization results.

| Method | Training dataset | MS$^2$ (RGB-T) | | VTD (RGB-T) | |
|---|---|---|---|---|---|
| | | EPE | F1 | EPE | F1 |
| CrossRAFT | Large-scale synthetic dataset | 6.21 | 70.20 | 9.86 | 89.33 |
| MINIMA | Large-scale synthetic dataset | 5.97 | 66.12 | 6.34 | 80.41 |
| NeMAR | MS$^2$ (RGB-T) | 19.25 | 99.80 | 25.29 | 99.99 |
| | VTD (RGB-T) | 20.05 | 99.76 | 23.43 | 96.89 |
| UMF-CMGR | MS$^2$ (RGB-T) | 18.84 | 99.78 | 32.06 | 99.99 |
| | VTD (RGB-T) | 25.78 | 99.99 | 28.05 | 99.13 |
| DCFlow (Ours) | MS$^2$ (RGB-T) | **3.46** | **35.89** | 6.65 | 76.25 |
| | VTD (RGB-T) | 5.70 | 68.97 | **3.65** | **48.49** |

Table 9: Generalization results from models trained on RGB-T or RGB-NIR image pairs to NIR-T.

| Training Modality | EPE | F1 |
|---|---|---|
| RGB-T | 10.20 | 86.23 |
| RGB-NIR | 27.66 | 99.99 |

**Analysis for low-performing modalities.** We note that the flow estimation accuracy of NIR-T modalities is lower than that of RGB-T. It is mainly caused by imperfect modality transfer. For instance, the NIR modality has a much narrower spectral range and only one channel, whereas RGB covers a wider spectrum with three channels. This makes transferring from NIR to thermal substantially more difficult than transferring from RGB, especially when perfectly aligned cross-modal pairs are not available for supervision. Fig. 11 compares modality transfer and flow estimation on RGB–T and NIR–T pairs from the same scenes. Compared with the transferred images from RGB, the transferred images from NIR show blurrier structures and less distinct edges in some regions, which result in larger alignment errors.

**Visualization of the training process.** We provide a visualization of the training process in Fig. 12. Fig. 12(a) shows the curves of four loss functions, where all losses exhibit stable decreases. Fig 12(b) presents the qualitative comparison of the modality transfer and flow estimation results at different training iterations. As the network trains, both the transferred images and estimated flow gradually improve, confirming that the proposed framework effectively learns cross-modal flow estimation over time.

**Qualitative comparison.** We present additional qualitative comparisons between DCFlow and previous methods, including NeMAR (Arar et al., 2020), UMF-CMGR (Wang et al., 2022), CrossRAFT (Zhou et al., 2022), and MINIMA (Ren et al., 2025), as shown in Fig. 13. It can be observed that DCFlow produces more accurate flow estimation results, whereas the other approaches exhibit noticeable mismatches.

**Efficiency comparison.** Table 10 presents the comparison in terms of inference time and memory usage. It can be seen that DCFlow achieves competitive runtime and memory usage, while maintaining strong accuracy.

## A.8 LIMITATIONS

While DCFlow significantly outperforms previous unsupervised approaches, a slight performance gap remains compared to supervised methods. In addition, DCFlow relies on a data synthesis pipeline to provide flow supervision from single-view images, which may increase training time. Despite these limitations, we believe DCFlow contributes to advancing unsupervised cross-modal flow estimation.

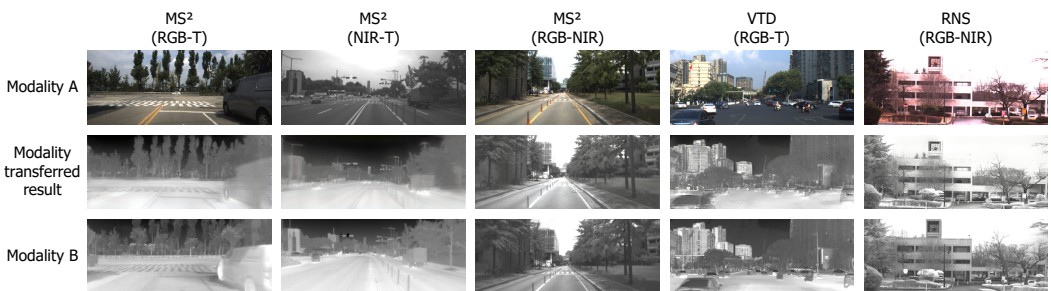

Figure 8: Visualization of modality transfer results across different datasets.

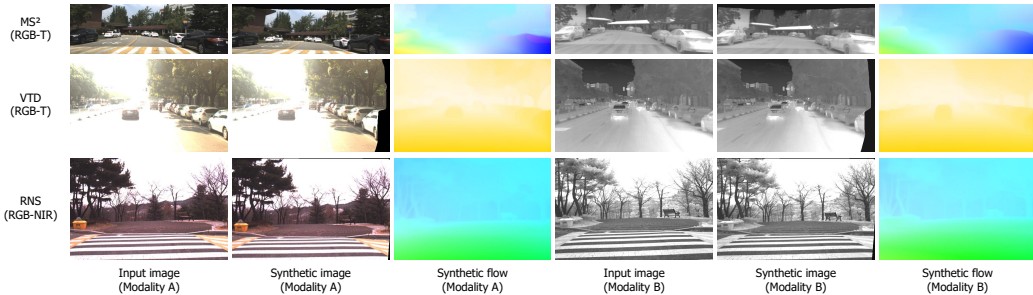

Figure 9: Visualization of synthetic flow data across different datasets.

# B  STATEMENT ON THE USE OF LLMS

We used large language models (LLMs) solely as a writing aid to improve grammar and wording. LLMs were not used for research ideas, experimental design, or data analysis. All text was written by the authors and carefully reviewed for accuracy and originality. The authors take full responsibility for the content of the manuscript.

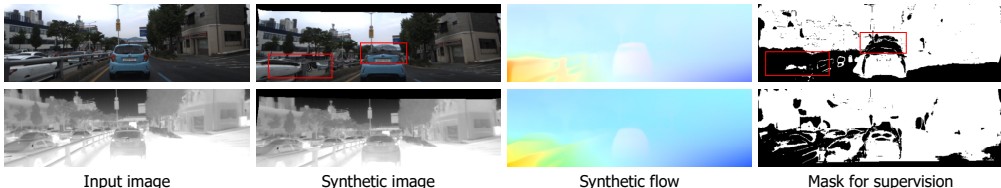

Figure 10: Visualization of the valid mask used for flow supervision on the MS$^2$ (RGB-T) dataset. Thanks to the photometric consistency checking and the outlier-robust loss, invisible areas and regions affected by severe artifacts are excluded from loss computation.

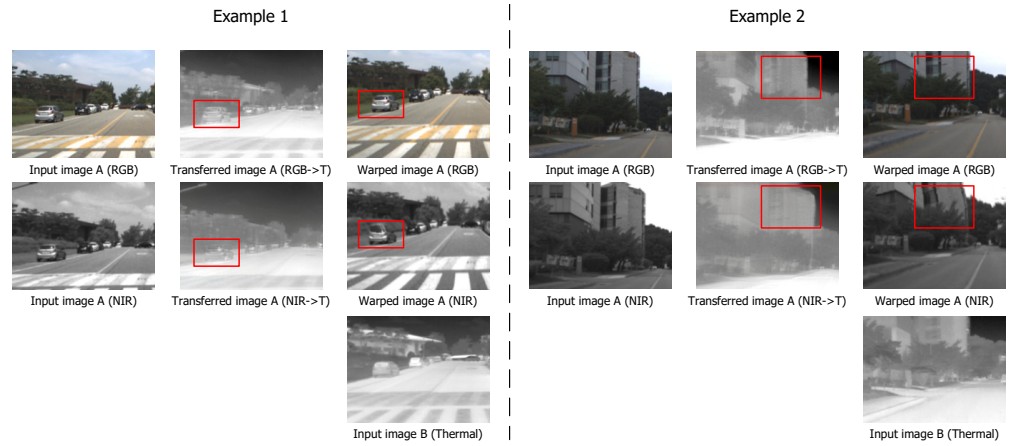

Figure 11: Comparison of modality transfer and flow estimation results for RGB–T and NIR–T pairs on the same scenes. For clarity, cropped patches of the full-resolution images are shown. The red boxes highlight regions where the modality transfer quality and the resulting flow estimation accuracy differ across modalities.

Table 10: Efficiency comparison of different approaches.

| Approach | RAFT | GMA | FlowFormer | SEA-RAFT | CrossRAFT | MINIMA | NeMAR | UMF-CMGR | DCFlow (Ours) |
|---|---|---|---|---|---|---|---|---|---|
| Inference time (ms) | 32.6 | 39.1 | 58.7 | 34.5 | 85.2 | 357.9 | 49.4 | 64.7 | 40.1 |
| Inference memory (MB) | 708 | 718 | 816 | 614 | 2250 | 7240 | 3056 | 2770 | 742 |

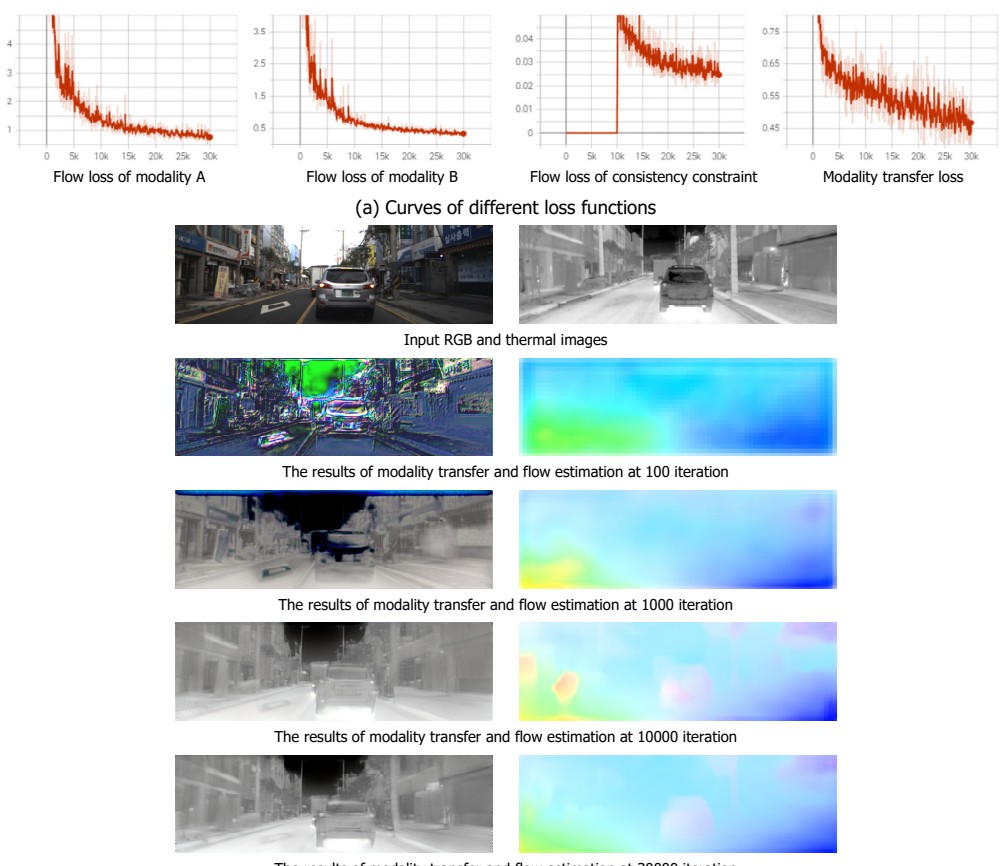

Figure 12: Visualization of the training process on the MS$^2$ (RGB-T) dataset. (a) Curves of loss functions used in DCFlow. All losses decrease with clear convergence as training progresses, indicating the effectiveness of our training strategies. (b) Visual examples of modality transfer and flow estimation at different training iterations. The results are shown at 100, 1000, 10000, and 30000 iterations.

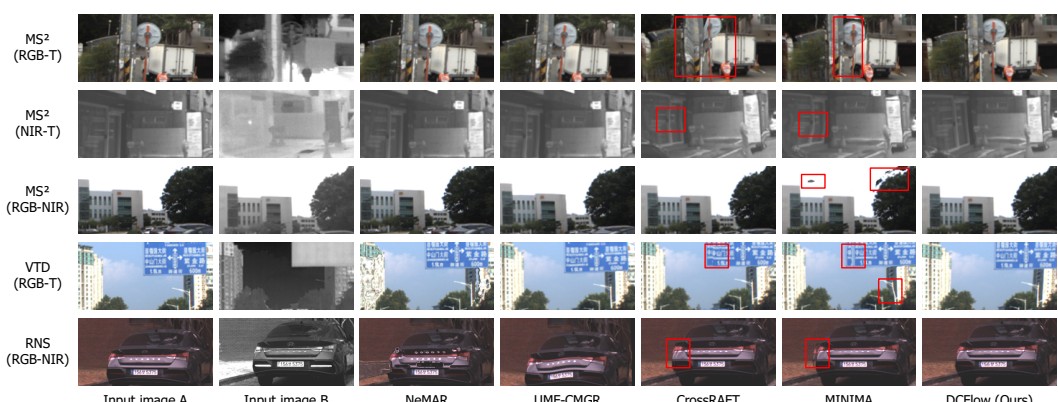

Figure 13: Qualitative comparison of DCFlow and other approaches. The first two columns show the input image pairs, and the remaining columns visualize the image from modality A warped using the estimated flow from each approach. For clarity, cropped patches of the full-resolution images are shown. The red boxes highlight the distortion regions.

