# OpenReview forum: "Rethinking Unsupervised Cross-modal Flow Estimation: Learning from Decoupled Optimization and Consistency Constraint"
_ICLR.cc/2026/Conference — ICLR 2026 Poster_

### Official Review · Reviewer_xLM8 · 2025-10-30

**Soundness:** 3
**Presentation:** 2
**Contribution:** 2
**Rating:** 6
**Confidence:** 4

**Summary:**

This paper proposes DCFlow, a decoupled cross-modal flow estimation framework that can jointly train its two components, a modality transfer network and a single-modality flow estimation, without requiring direct ground-truth supervision. Specifically, a geometry-aware data synthesis pipeline is used to synthesize new views as well as their flow to self-supervise the flow estimation network. Affine transformations are also applied to help refine both networks jointly for better cross-modal consistency. Experiment results show promising results for this method.

**Strengths:**

1. The paper is well-structured. It is great to have an overview of the proposed method before diving deep into each component.
2. The proposed method is simple and easy to understand. Each method design is explained in the text.
3. The presented results look good, although it may be better if the authors can show more results for more insights (detailed below in weaknesses and questions).

**Weaknesses:**

1. Academic writing can be better polished.
- Many references in the paper are cited using `\citet`, disrupting the flow of the text, which should be replaced with `\citep` for smoother readability.
- Some text also lacks clearance. For example, Line 083-085 says decoupled optimization reduces EPE by 72%, and Line 089-090 says the cross-modal consistency constraint results in a 28% improvement in EPE. This is confusing. Does it mean these two strategies jointly reduce EPE by 100%, so you basically have 0 EPE?
2. Eq (12) is not formatted correctly. "Argmin" stands for the argument values that can minimize the following expression, and adding these argument values does not make sense. Maybe the authors should put just one $\arg\min_{\phi\theta}$ outside the addition of these three loss components.
3. Table 2(c) is a bit weak because it lacks a baseline to compare. It is hard to tell whether the four network results are consistently improved compared to their original results without the proposed strategies in DCFlow.

**Questions:**

1. Since the two modalities are symmetric. Have the authors tried to switch the two modalities and see how the results may change? For example, if the current framework trains a modality transfer network from modality A to B and a flow estimation network that estimates flow in modality B. It is also possible to switch modality A and B (modality transfer from B to A and a flow estimation in A). It would be interesting to see how this will change the results, or at least the authors should justify why the current modality choices of A and B should be the best one.
2. Since the motion estimation network and the modality transfer network are optimized alternately, I assume the optimization should start with two decently working networks instead of training both from scratch. Could you discuss more on how to start this training?

---

> ### Author Response · Authors · 2025-11-19
> **Response to Reviewer xLM8 (1/2)**
>
> **W1. Academic writing.**
> **(1) Reference citation.**
> Thanks for your valuable suggestion. We have replaced the \citet with \citep in the revised manuscript.
> **(2) Some text also lacks clearance. For example, Line 083-085 says decoupled optimization reduces EPE by 72%, and Line 089-090 says the cross-modal consistency constraint results in a 28% improvement in EPE. This is confusing. Does it mean these two strategies jointly reduce EPE by 100%, so you basically have 0 EPE?**
> We apologize for the confusion. The “72% reduction” refers to the change from the appearance-based optimization baseline (the EPE of 21.23) to our decoupled optimization (the EPE of 5.80), and the “28% improvement” refers to the reduction from the EPE of 4.81 to 3.46 after adding the cross-modal consistency constraint. We have revised the manuscript (lines 85-92) to report the absolute EPE reductions directly instead of percentages for clarity.
>
> ---
>
> **W2. Formation of Eq.12.**
> Thanks for your valuable suggestion. We have revised the manuscript (lines 322-323) to correct this formulation.
>
> ---
>
> **W3. Table 2(c) is a bit weak because it lacks a baseline to compare. It is hard to tell whether the four network results are consistently improved compared to their original results without the proposed strategies in DCFlow.**
> To evaluate the effectiveness of our training strategy under different flow networks, we compare DCFlow with the appearance-based optimization baseline that relies solely on photometric losses. As shown in Table R1, the appearance-based baseline performs significantly worse, whereas DCFlow achieves superior results. We have added the above discussion in the revised manuscript (lines 397-403).
>
> *Table R1: Ablation on different flow networks.*
>
> |                               | RAFT           |                 | GMA            |                 | FlowFormer     |                 | SEA-RAFT       |                 |
> | :---------------------------- | :------------- | :-------------- | :------------- | :-------------- | :------------- | :-------------- | :------------- | :-------------- |
> |                               | EPE            | F1              | EPE            | F1              | EPE            | F1              | EPE            | F1              |
> | Appearance-based optimization | 21.23          | 98.45           | 24.27          | 97.39           | 29.22          | 99.43           | 23.93          | 98.66           |
> | DCFlow (Ours)                 | **3.46** | **35.89** | **4.13** | **48.04** | **3.66** | **37.97** | **3.57** | **38.63** |
>
> ---
>
> **Q1. Since the two modalities are symmetric. Have the authors tried to switch the two modalities and see how the results may change?**
> Since some modalities contain more information, transferring from richer to less informative modalities could make the learning process easier. We select the transfer direction based on the observation that modalities such as RGB and NIR contain richer texture and structure than thermal, making the modality transfer easier and more stable than the reverse direction. To verify this, we trained variants that reverse the direction of transfer. As shown in Table R2, transferring from RGB/NIR to thermal outperforms the reverse direction. The performance difference between RGB and NIR mapping is relatively small. We have added the above discussion in the revised manuscript (lines 463-468).
>
> *Table R2: Ablation on the modality transfer direction.*
>
> | Dataset | Direction | EPE  | F1    |
> |--------|-----------|------|-------|
> | **MS2 (RGB–T)** | RGB→T | 3.46 | 35.89 |
> |                | T→RGB | 3.68 | 44.11 |
> | **MS2 (NIR–T)** | NIR→T | 4.53 | 49.81 |
> |                | T→NIR | 4.80 | 60.71 |
> | **MS2 (RGB–NIR)** | RGB→NIR | 0.96 | 5.00 |
> |                  | NIR→RGB | 0.94 | 4.66 |

---

> ### Author Response · Authors · 2025-11-19
> **Response to Reviewer xLM8 (2/2)**
>
> **Q2. Since the motion estimation network and the modality transfer network are optimized alternately, I assume the optimization should start with two decently working networks instead of training both from scratch. Could you discuss more on how to start this training?**
>
> We apologize for the confusion. We'd like to clarify that both the flow estimation network and the modality transfer network are trained from scratch. The decoupled training strategy refers to the fact that each loss component only updates the parameters of its corresponding network. Although the two networks are not well-initialized at the beginning of training, our specific design ensures convergence towards the desired direction.
>
> First, the perceptual loss used for the modality transfer network is robust to small misalignments to some extent, allowing the modality transfer network to learn meaningful cross-modal mappings even when the early cross-modal flow estimations may be inaccurate. Second, the flow estimation network adopts a two-branch inter-modal supervision scheme using synthetic data from each modality. Following the principles of multi-task learning, it provides inherent regularization, and improves its robustness to imperfect modality transferred images. These two properties enable the networks to mutually improve each other during the early stages of training.
>
> We present a visualization of the overall training process in Fig.12 of the revised manuscript. As shown in Fig.12(a), all losses exhibit stable decreases during the training process. Fig.12(b) illustrates that both the modality transferred images and the estimated flows become increasingly accurate and stable as training progresses from 100 to 30000 iterations. This demonstrates that, despite starting from scratch, the specific design of DCFlow allows the two networks to converge effectively. We have added the above discussion in the revised manuscript (lines 949-955).

---

> > ### Author Response · Authors · 2025-11-27
> >
> > Dear Reviewer,
> >
> > Thank you sincerely for your thoughtful and constructive comments.
> >
> > With the discussion period nearing its end, we would like to confirm whether our responses have fully addressed your concerns. Should any further clarification be needed, we would be happy to provide it promptly.
> >
> > We truly appreciate your time and insights, and welcome any additional questions you may have.
> >
> > Best regards,
> >
> > The DCFlow authors

---

### Official Review · Reviewer_te33 · 2025-10-31

**Soundness:** 3
**Presentation:** 4
**Contribution:** 3
**Rating:** 6
**Confidence:** 3

**Summary:**

This paper proposes a cross-modal flow estimation framework (e.g., between RGB and thermal images) that does not require ground-truth optical flow across modalities. The key idea of the framework is to disentangle modality discrepancy and geometric misalignment. The architecture consists of a flow estimation network and a modality transfer network, which are trained both independently and jointly through a cross-modal consistency constraint that enhances cross-modal flow estimation. The proposed method is evaluated on a newly introduced dataset constructed from MS², VTD, and RNS, achieving superior performance compared to previous unsupervised cross-modal methods.

**Strengths:**

1. The proposed architecture effectively disentangles modality discrepancy and geometric misalignment, while the cross-modal consistency constraint strengthens the connection between modalities. This design is well-suited for scenarios where cross-modal ground-truth data is scarce and demonstrates effective results.

2. A comprehensive ablation study is conducted for each component, showing that every part of the proposed framework contributes to performance improvement on the evaluation dataset, thereby supporting the design choices.

**Weaknesses:**

1. Ambiguity in the use of the term unsupervised
- Although the term “unsupervised” is used throughout the paper, in my opinion its meaning differs from how it is conventionally used in the optical flow field. In this paper, the method is described as unsupervised because it does not require cross-modal optical flow annotations between modalities. However, each individual modality is trained using synthetic data with ground-truth optical flow annotations. Just as a optical flow model (e.g., RAFT)  trained on the FlyingChairs synthetic dataset is typically categorized as a supervised method, it is questionable whether the proposed pipeline, which is trained on synthetic data with GT flow, should be classified as unsupervised. It would be better to use a more precise term that explicitly conveys the idea of training without cross-modal ground-truth flow, rather than referring to the method as completely unsupervised.

2.  Novelty of outlier-robust loss and geometry-aware data synthesis pipeline
- In Lines 81–82, the paper states “we further design an outlier-robust loss”, and in Line 253, it mentions “we propose a geometry-aware data synthesis pipeline with an outlier-robust loss.” However, both ideas appear to overlap with previous works. For the outlier-robust loss, several prior studies—such as WaterMono [a1], MapAnything [a2], and Regularizing Nighttime Weirdness [a3] already adopt a similar strategy by excluding the top percentile of high-error pixels during loss computation. Likewise, the idea of a geometry-aware data synthesis pipeline has been previously explored in [a4] and [a5]. It would be better for the authors to properly cite the relevant prior works to clarify the originality and positioning of these components.

[a1] Ding et al., “WaterMono: Teacher-Guided Anomaly Masking and Enhancement Boosting for Robust Underwater Self-Supervised Monocular Depth Estimation”, IEEE Transactions on Instrumentation and Measurement 2025.

[a2] Keetha et al., “MapAnything: Universal Feed-Forward Metric 3D Reconstruction”, arXiv 2025.

[a3] Wang et al, “Regularizing nighttime weirdness: Efficient self-supervised monocular depth estimation in the dark.”, ICCV 2021.

[a4] Aleotti et al., “Learning optical flow from still images”, CVPR 2021.

[a5] Han, Yunhui, et al., “RealFlow: EM-based Realistic Optical Flow Dataset Generation from Videos”, ECCV 2022.

3. The proposed experiments are conducted on MS², VTD, and RNS datasets, using 80% of frames per video for training and 20% for testing. However, this evaluation setup may be insufficient to assess the model’s generalization capability, since both training and testing samples originate from the same video distribution. It would be more convincing if the authors could report results under a cross-dataset evaluation setting, where the training and evaluation datasets differ, to better demonstrate generalization performance.

4. It would be beneficial if the paper provided a visualization of the training process for the proposed flow estimation network and modality transfer module.

**Questions:**

1. In Section 3.4, could the authors include more details on how the transformed flow is obtained? In my opinion, while this operation seems feasible, it likely involves non-trivial preprocessing or transformation steps. It would be helpful to include a brief explanation or illustration of this process in the supplementary material.

**Details Of Ethics Concerns:**

This work focuses on optical flow estimation and uses publicly available datasets, so there appear to be no ethical or licensing concerns associated with the data or methodology.

---

> ### Author Response · Authors · 2025-11-19
> **Response to Reviewer te33 (1/2)**
>
> **W1. Ambiguity in the use of the term unsupervised.**
> Thanks for your valuable suggestion. We agree that our DCFlow differs from traditional unsupervised approaches that rely solely on photometric loss. Our original use of "unsupervised" was intended to indicate that our framework does not require cross-modal ground-truth flow, which is in line with other unsupervised approaches. However, we acknowledge that the geometry-aware data synthesis pipeline introduces flow supervision in a self-supervised manner. To improve clarity and precision, we have revised the manuscript as follows:
> - We have adopted the term "unsupervised" to describe the problem setting (no cross-modal ground-truth flow).
> - We have adopted the term "self-supervised" or "training without cross-modal ground-truth flow" to describe the specific training strategy of DCFlow.
>
> ---
>
> **W2. Novelty of outlier-robust loss and geometry-aware data synthesis pipeline.**
> We agree that both the outlier-robust loss and geometry-aware data synthesis pipeline are not entirely new separately. Prior works have explored similar ideas in different contexts. However, we note that the novelty of our approach lies in integrating these two techniques into the decoupled optimization pipeline, thus enabling mono-modal flow supervision to benefit cross-modal flow estimation.
> We have cited and discussed the relevant prior works, and refined the statements to clearly reflect the originality of our method in the revised manuscript (lines 43-48, 132-134, 282-285).
>
> ---
>
> **W3. It would be more convincing if the authors could report results under a cross-dataset evaluation setting, where the training and evaluation datasets differ, to better demonstrate generalization performance.**
> We present the cross-dataset generalization results between MS2 (RGB–T) and VTD (RGB–T) in Table R1. When trained on one dataset and evaluated on another unseen dataset, DCFlow achieves better performance than existing unsupervised approaches trained directly on the target dataset, as reported in Table 4 of the manuscript. These results demonstrate the generalization ability of DCFlow. We have added the above discussion in the revised manuscript (lines 891-895).
>
> *Table R1: Cross-dataset generalization results.*
>
> | Training Dataset / Test Dataset | MS2 (RGB-T) |       | VTD (RGB-T) |       |
> | :-----------------------------: | :---------- | :---- | :---------- | :---- |
> |                                | EPE         | F1    | EPE         | F1    |
> |           MS2 (RGB-T)           | 3.46        | 35.89 | 6.65        | 76.25 |
> |           VTD (RGB-T)           | 5.70        | 68.97 | 3.65        | 48.49 |
>
> ---
>
> **W4. It would be beneficial if the paper provided a visualization of the training process for the proposed flow estimation network and modality transfer module.**
> We present the visualization of the training process on the MS2(RGB-T) dataset in Fig.12 of the revised manuscript. Fig.12(a) shows the curves of four loss functions. All losses decrease with clear convergence as training progresses, indicating the effectiveness of our training strategies. Fig.12(b) presents the qualitative comparison of the modality transfer and flow estimation results at different training iterations. As the network trains, both the transferred images and estimated flow gradually improve, confirming that the proposed framework effectively learns cross-modal flow estimation over time. We have added the above discussion in the revised manuscript (lines 937-942).

---

> > ### Author Response · Authors · 2025-11-19
> > **Response to Reviewer te33 (2/2)**
> >
> > **Q1. In Section 3.4, could the authors include more details on how the transformed flow is obtained?**
> > Given a cross-modal image pair $(\mathbf{I}\_\mathrm{A}, \mathbf{I}\_\mathrm{B})$ and the predicted flow $\mathbf{F}\_\mathrm{B2A}$, we apply the same random affine transformation $\mathcal{A}(\mathbf{x}) = \mathbf{A}\mathbf{x} + \mathbf{t}$ to both images, where $\mathbf{A}$ encodes rotation and scaling, and $\mathbf{t}$ is the translation. We denote the original flow as $\mathbf{F}\_\mathrm{B2A}(\mathbf{x}\_\mathrm{B}) = \mathbf{x}\_\mathrm{A} - \mathbf{x}\_\mathrm{B}$. After transformation, the corresponding points are given by $\tilde{\mathbf{x}}\_\mathrm{B} = \mathcal{A}(\mathbf{x}\_\mathrm{B})$ and $ \tilde{\mathbf{x}}\_\mathrm{A} = \mathcal{A}(\mathbf{x}\_\mathrm{A})$. The transformed flow can be formulated as $ \tilde{\mathbf{F}}\_\mathrm{B2A}^{*}(\tilde{\mathbf{x}}\_\mathrm{B}) = \tilde{\mathbf{x}}\_\mathrm{A} - \tilde{\mathbf{x}}\_\mathrm{B} = \mathbf{A}(\mathbf{x}\_\mathrm{A} - \mathbf{x}\_\mathrm{B}) = \mathbf{A} \mathbf{F}\_\mathrm{B2A}(\mathbf{x}\_\mathrm{B})$. We note that only the linear part $\mathbf{A}$ affects the flow, while the translation term $\mathbf{t}$ cancels out. In practice, for each $\tilde{\mathbf{x}}\_\mathrm{B}$, we sample the corresponding $\mathbf{F}\_\mathrm{B2A}(\mathbf{x}\_\mathrm{B})$ via bilinear interpolation and then multiply it by $\mathbf{A}$ to obtain the transformed flow. We have added the above discussion in the revised manuscript (lines 734-745).

---

> ### Comment · Reviewer_te33 · 2025-11-25
>
> Thank you for your response. Please find my clarifications below.
>
> **Weakness 1. Ambiguity in the use of the term “unsupervised.”**
> Thank you for correcting the terminology. It would be helpful if the abstract or introduction included a brief clarification of what is meant by “self-supervised” or “unsupervised,” for example by explaining that the method does not require cross-modal optical-flow annotations. Currently, the meaning of self-supervised or unsupervised only becomes clear in the later sections of the paper.
> **Weakness 2. Novelty of the outlier-robust loss and the geometry-aware data synthesis pipeline."**
> In my understanding, generating multi-spectrum synthetic data requires the multi-spectrum images to be aligned with the corresponding RGB images, since the pipeline relies on depth maps predicted from the RGB domain. Because of this dependency, the approach does not seem substantially different from prior works. The outlier-robust loss also appears similar to existing literature. Nonetheless, I still value the effectiveness and overall design of the proposed training pipeline.
> **Weakness 3. Experimental comparison."**
> Thank you for providing the experiments. If possible, it would be better to additionally report the performance of competitor methods under the same evaluation setting.
> **Weakness 4."**
> Thank you! this has been addressed.

---

> ### Author Response · Authors · 2025-11-27
> **Response to Reviewer te33**
>
> Thanks for your valuable feedback. We address your concerns below point by point, and look forward to your further comments.
>
> ---
>
> **W1. Ambiguity in the use of the term "unsupervised".**
> Thank you for pointing this out. We have revised the introduction to clarify the meanings of "unsupervised" and "self-supervised" as follows:
> - Unsupervised: **Unsupervised** cross-modal flow estimation, which does not require ground-truth cross-modal flow annotations for training, has attracted increasing attention. (lines 33-35)
> - Self-supervised: DCFlow learns in a fully **self-supervised** manner by generating flow supervision through data synthesis, using only unaligned cross-modal images. (lines 76-77)
>
> ---
>
> **W2. Novelty of outlier-robust loss and geometry-aware data synthesis pipeline.**
> Thank you for your insightful comment and for acknowledging the overall design of our training pipeline. We would like to first clarify that DCFlow **does not require multi-modal images to be aligned with the corresponding RGB images**, and we **do not generate multi-modal synthetic data using depth maps predicted from RGB**. Instead, depth estimation and geometry-aware synthesis are performed independently for each modality, which enables DCFlow to work directly with unaligned cross-modal image pairs.
>
> Moreover, the key contribution of DCFlow lies in its decoupled optimization strategy, which separates the overall problem into modality transfer and flow estimation, each equipped with its own supervision. This design allows single-modal flow data to effectively support cross-modal flow estimation, and overcomes the limitations of existing unsupervised approaches that depend solely on appearance similarity. Under this decoupled optimization framework, the outlier-robust loss is adopted to improve the reliability of flow supervision by mitigating the inevitable artifacts introduced during geometry-aware synthesis, which in turn further enhances the performance of the decoupled training strategy.
>
> ---
>
> **W3.  Experimental comparison.**
> Thank you for the helpful suggestion. We report the cross-dataset generalization results of all large-scale pretrained and unsupervised baselines in Table R1. Compared with unsupervised baselines such as NeMAR and UMF-CMGR, DCFlow achieves significantly better cross-dataset performance, and even surpasses these approaches when they are trained directly on the target dataset. This highlights the advantage of our framework over previous unsupervised approaches that rely solely on appearance-based supervision. Moreover, although large-scale pretrained approaches use substantially larger synthetic datasets with ground-truth flow, DCFlow achieves comparable or better generalization in the cross-dataset evaluation setting, while requiring far less training data. We have added this discussion to the revised manuscript (lines 892–899).
>
> *Table R1: Cross-dataset generalization results.*
>
> | Method | Training dataset | Test dataset |  |  |  |
> |:---:|:---:|:---:|:---:|:---:|:---:|
> |  |  | MS2 (RGB-T) |  | VTD (RGB-T) |  |
> |  |  | EPE | F1 | EPE | F1 |
> | CrossRAFT | Large-scale synthetic dataset | 6.21 | 70.20 | 9.86 | 89.33 |
> | MINIMA | Large-scale synthetic dataset | 5.97 | 66.12 | 6.34 | 80.41 |
> | NeMAR | MS2 (RGB-T) | 19.25 | 99.80 | 25.29 | 99.99 |
> |  | VTD (RGB-T) | 20.05 | 99.76 | 23.43 | 96.89 |
> | UMF-CMGR | MS2 (RGB-T) | 18.84 | 99.78 | 32.06 | 99.99 |
> |  | VTD (RGB-T) | 25.78 | 99.99 | 28.05 | 99.13 |
> | DCFlow (Ours) | MS2 (RGB-T) | 3.46 | 35.89 | 6.65 | 76.25 |
> |  | VTD (RGB-T) | 5.70 | 68.97 | 3.65 | 48.49 |

---

### Official Review · Reviewer_NaNT · 2025-11-03

**Soundness:** 3
**Presentation:** 2
**Contribution:** 3
**Rating:** 6
**Confidence:** 3

**Summary:**

The paper proposes DCFlow, an unsupervised framework for cross-modal flow that trains a modality-transfer network and a flow network with a decoupled scheme plus a cross-modal consistency objective. The flow net learns from explicit mono-modal supervision generated by a geometry-aware single-image synthesis pipeline with an outlier-robust loss; the transfer net learns via a perceptual loss while flow is frozen; and a consistency loss enforces equivariance under known spatial transforms to tie the two together.

**Strengths:**

(1) A decoupled optimization recipe that stabilizes training by supervising the flow net on synthetic mono-modal pairs while separately optimizing the transfer net, addressing modality discrepancy and geometric misalignment distinctly.

(2) A geometry-aware single-image synthesis pipeline + top-τ% trimming that yields robust, dense motion supervision without cross-modal labels.

(3) A cross-modal consistency constraint that jointly optimizes the two nets under known spatial transforms.

(4) A repurposed benchmark over MS2/VTD/RNS with RGB/NIR/Thermal pairs and LiDAR-derived sparse flow.

**Weaknesses:**

1. The paper jumps quickly from identifying limitations of existing appearance-based methods to presenting its full “decoupled + consistency” solution. The reasoning that connects the problem to this specific design is underdeveloped and feels somewhat ad hoc.

2. It seems that several key components, such as the fixed top-τ% robust loss, small affine ranges in the consistency constraint, and reliance on monocular depth synthesis, are justified only empirically with little or none sensitivity or ablation analysis.

3. While quantitative results are strong, the paper can be refined by introducing an analysis of efficiency, runtime, and generalization to unseen domains or sensor types.

**Questions:**

(1) How is monocular depth obtained for non-RGB modalities (Thermal, NIR), and how reliable is it for generating supervision?

(2) Why was the top-τ% value fixed at 20%, and how sensitive are results to this choice or to alternative robust loss functions?

---

> ### Author Response · Authors · 2025-11-19
> **Response to Reviewer NaNT (1/3)**
>
> **W1. The paper jumps quickly from identifying limitations of existing appearance-based methods to presenting its full “decoupled + consistency” solution. The reasoning that connects the problem to this specific design is underdeveloped and feels somewhat ad hoc.**
> We attribute the limitations of existing appearance-based approaches to relying solely on appearance similarity for flow estimation. This solution couples the tasks of modality transfer and motion estimation, and is suboptimal in textureless regions or repetitive structures. Motivated by these limitations, we believe that a potential improvement is to introduce motion supervision without using the cross-modal ground-truth flow. To achieve this goal, we decouple the overall task into modality transfer and mono-modal flow estimation, thus enabling mono-modal flow supervision to benefit cross-modal flow estimation. Furthermore, to ensure effective interaction between the two tasks, we introduce the cross-modal consistency constraint. This constraint facilitates collaboration between the networks, leading to improved performance. By combining decoupled training with the consistency constraint, we form our full solution. We have revised the manuscript (lines 160-192) to better illustrate the reasoning of our design.
>
> ---
>
> **W2. It seems that several key components, such as the fixed top-τ% robust loss, small affine ranges in the consistency constraint, and reliance on monocular depth synthesis, are justified only empirically with little or none sensitivity or ablation analysis.**
> We conduct additional sensitivity and ablation experiments to evaluate the influence on performance of the top τ% robust loss, the affine ranges in the consistency constraint, and the reliance on monocular depth synthesis.
> **(1) Top-τ% robust loss.**
> We vary τ from 10 to 40, and list the results in Table R1. When τ is too small, the filtering of undesired regions becomes insufficient, and noisy pixels still contribute to supervision. When τ is too large, too many pixels are removed, and the supervision signal becomes weak. We choose τ as 20 to provide a good balance between removing unreliable regions and keeping enough valid information for supervision.
>
> *Table R1: Ablation on the value of τ.*
>
> |           τ           |      EPE      |       F1       |
> | :---------------------: | :------------: | :-------------: |
> | w/o outlier-robust loss |      5.80      |      57.18      |
> |           10           |      5.40      |      55.59      |
> |        20 (Ours)        | **4.81** |      51.39      |
> |           30           |      5.01      | **50.22** |
> |           40           |      5.59      |      58.29      |
>
> **(2) Affine ranges in the consistency constraint.**
> We evaluate small, medium, and large affine perturbations for consistency constraint, and list the results in Table R2. It can be seen that small and medium perturbations already introduce sufficient geometric variation for the consistency constraint, while excessively large perturbations may create unrealistic deformations that harm the supervision quality. We choose medium perturbations as default.
>
> *Table R2: Ablation on the affine ranges.*
>
> |        Affine range        | Rotation | Translation |   Scaling   |      EPE      |       F1       |
> | :------------------------: | :------: | :---------: | :---------: | :------------: | :-------------: |
> | w/o consistency constraint |    -    |      -      |      -      |      4.81      |      51.39      |
> |           Small           | [-1,+1] |  [-12,+12]  | [0.98,1.02] |      3.61      |      36.08      |
> |       Medium (Ours)       | [-3,+3] |  [-24,+24]  | [0.95,1.05] | **3.46** | **35.89** |
> |           Large           | [-5,+5] |  [-36,+36]  | [0.92,1.08] |      3.93      |      46.37      |

---

> ### Author Response · Authors · 2025-11-19
> **Response to Reviewer NaNT (2/3)**
>
> **(3) Monocular depth synthesis.**
> The monocular depth estimation is needed to obtain the relative distance relationships for constructing geometry-aware flow synthesis. However, DCFlow does not rely on highly accurate depth estimation, and it is designed to be robust even when the depth is inaccurate or imperfect. Both the photometric consistency checking and the outlier-robust loss exclude unreliable regions from flow supervision.
> To verify the robustness, we conduct stress testing by adding Gaussian noise, scaling noise, and edge blurring to the depth maps produced by UniDepth [A1]. As shown in Table R3, DCFlow shows only slight performance drop even when the depth maps are heavily corrupted, and it still performs favorably compared with existing unsupervised baselines. We also replace UniDepth with Metric3D v2 [A2] and Depth Anything v2 [A3]. As shown in Table R4, DCFlow achieves consistent performance across these depth networks, which further shows that it does not depend on a specific depth model.
>
> *Table R3: Stress-testing under inaccurate depth estimation.*
>
> |   Depth quality   | σ of Gaussian noise | σ of scaling noise | Kernel size of blurring |      EPE      |       F1       |
> | :---------------: | :------------------: | :-----------------: | :---------------------: | :------------: | :-------------: |
> |  Network output  |          -          |          -          |            -            | **3.46** | **35.89** |
> | Small degradation |         0.1         |         0.1         |            7            |      3.81      |      40.66      |
> | High degradation |         0.2         |         0.2         |           14           |      4.00      |      49.08      |
>
> *Table R4: Comparison of different monocular depth estimation networks.*
>
> |   Depth network   |      EPE      |       F1       |
> | :---------------: | :------------: | :-------------: |
> |    Metric3D v2    |      3.74      |      39.41      |
> | Depth Anything v2 | **3.45** | **34.81** |
> |     UniDepth     |      3.46      |      35.89      |
>
> We have added the above discussion in the revised manuscript (lines 425-458, lines 857-888).
>
> ---
>
> **W3. While quantitative results are strong, the paper can be refined by introducing an analysis of efficiency, runtime, and generalization to unseen domains or sensor types.**
> **(1) Efficiency and runtime.**
> Table R5 presents the comparison in terms of inference time and memory usage. It can be seen that DCFlow achieves competitive runtime and memory usage, while maintaining strong accuracy.
>
> *Table R5: Efficiency comparison of different approaches.*
>
> |                      | RAFT | GMA | FlowFormer | SEA-RAFT | CrossRAFT | MINIMA | NeMAR | UMF-CMGR | DCFlow (Ours) |
> | :-------------------: | :--: | :--: | :--------: | :------: | :-------: | :----: | :---: | :------: | :-----------: |
> |  Inference time (ms)  | 32.6 | 39.1 |    58.7    |   34.5   |   85.2   | 357.9 | 49.4 |   64.7   |     40.1     |
> | Inference memory (MB) | 708 | 718 |    816    |   614   |   2250   |  7240  | 3056 |   2770   |      742      |
>
> **(2) Generalization to unseen domains.**
> Table R6 reports the cross-dataset generalization results between MS2 (RGB–T) and VTD (RGB–T). When trained on one dataset and evaluated on another unseen dataset, DCFlow achieves better performance than existing unsupervised approaches trained directly on the target dataset, as reported in Table 4 of the manuscript. These results demonstrate the generalization ability of DCFlow.
>
> *Table R6: Cross-dataset generalization results.*
>
> | Training Dataset / Test Dataset | MS2 (RGB-T) |       | VTD (RGB-T) |       |
> | :-----------------------------: | :---------- | :---- | :---------- | :---- |
> |                                | EPE         | F1    | EPE         | F1    |
> |           MS2 (RGB-T)           | 3.46        | 35.89 | 6.65        | 76.25 |
> |           VTD (RGB-T)           | 5.70        | 68.97 | 3.65        | 48.49 |
>
> **(3) Generalization to unseen sensor types.**
> We further evaluate the cross-modal generalization ability of DCFlow by testing models trained on MS2(RGB–T) and MS2(RGB–NIR) on the unseen MS2(NIR–T) modality. As shown in Table R7, the model trained on RGB–T achieves reasonable performance on NIR–T, whereas the model trained on RGB–NIR fails to generalize. This is because the modality gap between RGB–T and NIR–T is relatively closer, making the transfer from RGB–T to NIR–T more feasible. In contrast, the gap between RGB–NIR and NIR–T is substantially larger, which leads to poor generalization.
>
> *Table R7: Generalization results from models trained on RGB-T or RGB-NIR image pairs to NIR-T.*
>
> | Training Modality | EPE | F1 |
> | :---------------: | :--: | :-:|
> |       RGB-T       | 10.20 | 86.23 |
> |      RGB-NIR      | 27.66 | 99.99 |
>
> We have added the above discussion in the revised manuscript (lines 892-905, lines 960-963).

---

> ### Author Response · Authors · 2025-11-19
> **Response to Reviewer NaNT (3/3)**
>
> **Q1. How is monocular depth obtained for non-RGB modalities (Thermal, NIR), and how reliable is it for generating supervision?**
> **(1) How is monocular depth obtained for non-RGB modalities.**
> The monocular depth for NIR and thermal images is obtained by directly applying depth foundation models pretrained on the RGB domain in a zero-shot manner.
> **(2) How reliable is it for generating supervision.**
> Although these depth models are trained on RGB data, prior works [A4, A5] show that monocular depth estimation mainly relies on geometry cues such as perspective geometry, occlusion boundaries, and texture gradients, which are largely modality invariant. Moreover, these depth foundation models [A1, A2, A3] are trained on diverse datasets with strong augmentations such as brightness, contrast, and hue changes, which further enhance their generalization ability to unseen modalities. As a result, models such as UniDepth and Depth Anything generalize reasonably well to NIR and thermal inputs.
> More importantly, DCFlow does not rely on depth being highly accurate. Depth is only used as an intermediate variable to construct self-consistent geometry-aware synthetic flows. Since both the warped image and the synthetic flow are generated from the same depth prediction, the resulting synthetic flow label can reliably represent the motion between the source image and the warped image. In addition, the photometric consistency checking and the outlier-robust loss filter out invisible or artifact prone regions, ensuring that supervision focuses on pixels with reliable motion and appearance.
> To further verify reliability, we conduct stress testing by degrading the depth maps predicted by UniDepth. As shown in Table R3, DCFlow shows only slight performance drop even when the depth maps are heavily corrupted, and it still performs favorably compared with existing unsupervised baselines, indicating that highly accurate depth is not a strict requirement. We also replace UniDepth with Metric3D v2 and Depth Anything v2. As shown in Table R4, DCFlow achieves consistent performance across all depth networks.
> We have added the above discussion in the revised manuscript (lines 448-458).
>
> ---
>
> **Q2. Why was the top-τ% value fixed at 20%, and how sensitive are results to this choice or to alternative robust loss functions?**
> **(1) Ablation of top-τ% value.**
> We present the ablation of the top τ percent value in Table R1. All settings are competitive with or better than the baseline without outlier removal. We choose τ equal to 20 because it provides a good balance between filtering unreliable pixels and retaining enough valid information for supervision.
> **(2) Comparison with alternative robust loss functions.**
> We compare our outlier-robust loss with the Charbonnier loss and the Mixture of Laplace loss used in SEA-RAFT [A6]. As shown in Table R8, both alternatives lead to inferior results, while our outlier-robust loss achieves the best performance by more effectively suppressing high error outliers.
>
> *Table R8: Comparison of different robust loss functions.*
>
> |         Loss type         |      EPE      |       F1       |
> | :------------------------: | :------------: | :-------------: |
> |     L1 loss (baseline)     |      5.80      |      57.18      |
> |      Charbonnier loss      |      6.02      |      63.62      |
> |  Mixture-of-Laplace loss  |      5.94      |      56.21      |
> | Outlier-robust loss (Ours) | **4.81** | **51.39** |
>
> We have added the above discussion in the revised manuscript (lines 857-888).
>
> ---
>
> **References:**
> [A1] UniDepth: Universal Monocular Metric Depth Estimation. CVPR, 2024.
> [A2] Metric3D v2: A Versatile Monocular Geometric Foundation Model for Zero-shot Metric Depth and Surface Normal Estimation. TPAMI, 2024.
> [A3] Depth Anything V2. NeurIPS, 2024.
> [A4] Deep Depth Estimation from Thermal Image. CVPR, 2023.
> [A5] Deep Depth Estimation from Thermal Image: Dataset, Benchmark, and Challenges. arXiv, 2025.
> [A6] SEA-RAFT: Simple, Efficient, Accurate RAFT for Optical Flow. ECCV, 2024.

---

> > ### Author Response · Authors · 2025-11-27
> >
> > Dear Reviewer,
> >
> > Thank you sincerely for your thoughtful and constructive comments.
> >
> > With the discussion period nearing its end, we would like to confirm whether our responses have fully addressed your concerns. Should any further clarification be needed, we would be happy to provide it promptly.
> >
> > We truly appreciate your time and insights, and welcome any additional questions you may have.
> >
> > Best regards,
> >
> > The DCFlow authors

---

### Official Review · Reviewer_PNVY · 2025-11-04

**Soundness:** 3
**Presentation:** 3
**Contribution:** 3
**Rating:** 6
**Confidence:** 2

**Summary:**

This work introduces a method for cross-modal optical flow estimation. It claims that different modalities of data have their own applicability. For example, during autonomous driving, different modalities of data may be required for day and night. In such cases, there may be issues such as misalignment of different modalities of data, making cross-modal optical flow estimation relatively difficult. Specifically, it proposes DCFlow, which significantly improves the model performance in an unsupervised manner based on the decoupled optimization strategy and synthetic supervision data.

**Strengths:**

- DCFlow clearly addresses a key limitation in present unsupervised cross-modal flow methods by introducing direct, geometry-aware supervision and robust optimization.
- By conducting separate training with single-modal optical flow supervision and modal transfer supervision, the problem of task coupling in traditional methods is resolved.
- The experimental section is thorough, systematically demonstrating the effect of each technical choice.
- This method circumvents the need for supervised data by using synthetic data, and it is effective and sound.

**Weaknesses:**

- The approach relies on pretrained monocular depth (Sec 3.3, Fig. 3) to generate synthetic flows. However, there is minimal discussion of the risk or variability this introduces. How does DCFlow handle inaccurate or biased depth prediction in unseen modalities, e.g., thermal-NIR? What happens if the depth model underperforms? This is especially noteworthy given the visual artifacts in certain synthetic example patches (Fig. 4/9). The ablation on the synthetic pipeline is strong, but further stress-testing (e.g., on failure cases with poor depth) is absent.

**Questions:**

- How robust is DCFlow to failures in monocular depth estimation for the geometry-aware synthetic pipeline?
- For low-performing modalities/gaps (e.g., MS^2 NIR-T), can you provide qualitative failure analysis or error breakdown by source (e.g., transfer vs. flow estimation vs. data synthesis)?

---

> ### Author Response · Authors · 2025-11-19
> **Response to Reviewer PNVY (1/2)**
>
> **W1. The approach relies on pretrained monocular depth (Sec 3.3, Fig. 3) to generate synthetic flows. However, there is minimal discussion of the risk or variability this introduces. How does DCFlow handle inaccurate or biased depth prediction in unseen modalities, e.g., thermal-NIR? What happens if the depth model underperforms? This is especially noteworthy given the visual artifacts in certain synthetic example patches (Fig. 4/9). The ablation on the synthetic pipeline is strong, but further stress-testing (e.g., on failure cases with poor depth) is absent.**
> Thanks for your valuable question. In general, the estimated depth serves only as an intermediate variable for flow data synthesis. The photometric consistency checking and the outlier-robust loss enable us to exclude undesired regions from flow supervision. We discuss this from two aspects.
> **(1) The robustness of DCFlow under inaccurate or biased depth prediction.**
> Although the depth prediction may not be perfect, it is sufficient to capture the relative distance relationships between different regions of an image. Since both the warped image and the corresponding synthetic flow are computed using the same depth prediction, the synthetic flow label can still reliably represent the motion between the source image and the warped image. In addition, the photometric consistency checking and the outlier-robust loss filter out invisible areas and regions with artifacts, which ensures that the supervision focuses on pixels with reliable motion and appearance. Moreover, the cross-modal consistency constraint provides an additional source of flow supervision that does not rely on monocular depth. To demonstrate the robustness of DCFlow under inaccurate depth, we add degradations to the depth map estimated by UniDepth [A1] for flow synthesis, including Gaussian noise, scaling noise, and edge blurring. As shown in Table R1, DCFlow shows only slight performance drop even when the depth maps are heavily corrupted, and it still performs favorably compared with existing unsupervised baselines.
>
> *Table R1: Stress-testing under inaccurate depth estimation.*
>
> |   Depth quality   | σ of Gaussian noise | σ of scaling noise | Kernel size of blurring |      EPE      |       F1       |
> | :---------------: | :------------------: | :-----------------: | :---------------------: | :------------: | :-------------: |
> |  Network output  |          -          |          -          |            -            | **3.46** | **35.89** |
> | Small degradation |         0.1         |         0.1         |            7            |      3.81      |      40.66      |
> | High degradation |         0.2         |         0.2         |           14           |      4.00      |      49.08      |
>
> We further present examples of the valid mask in Fig.10 of the revised manuscript. Although imperfect depth may introduce visual artifacts in the synthetic images, the photometric consistency checking and the outlier-robust loss effectively remove these unreliable regions from loss computation, which ensures high-quality supervision for the flow network.
>
> **(2) The generalization of monocular depth networks for unseen modalities.**
> We observe that monocular depth models trained on the RGB domain show good generalization to unseen modalities like NIR and thermal. This can likely be attributed to the fact that monocular depth estimation primarily relies on perspective geometry and cues such as object size, occlusion, and texture gradients, which are shared across different modalities [A2, A3]. In addition, these depth models are trained on diverse datasets that cover a wide range of environments, and the training includes strong augmentations such as brightness, contrast, and hue changes [A1, A4, A5]. These strategies further enhance the generalization ability of depth models for unseen modalities. We replace UniDepth [A1] with Depth Anything v2 [A4] and Metric3D v2 [A5] for flow synthesis, and the results in Table R2 show that DCFlow consistently achieves strong performance across different depth networks.
>
> *Table R2: Comparison of different monocular depth estimation networks.*
>
> |   Depth network   |      EPE      |       F1       |
> | :---------------: | :------------: | :-------------: |
> |    Metric3D v2    |      3.74      |      39.41      |
> | Depth Anything v2 | **3.45** | **34.81** |
> |     UniDepth     |      3.46      |      35.89      |
>
> We have added the above discussion in the revised manuscript (lines 425-458, lines 914-917).

---

> ### Author Response · Authors · 2025-11-19
> **Response to Reviewer PNVY (2/2)**
>
> **Q1. How robust is DCFlow to failures in monocular depth estimation for the geometry-aware synthetic pipeline?**
> To explore the robustness of DCFlow under failures in monocular depth estimation, we add several degradations to the depth map estimated by UniDepth. As shown in Table R1, DCFlow shows only slight performance drop even when the depth maps are heavily corrupted, and still achieves better performance than the existing unsupervised approaches. We have added the above discussion in the revised manuscript (lines 425-447).
>
> ---
>
> **Q2. For low-performing modalities/gaps (e.g., MS^2 NIR-T), can you provide qualitative failure analysis or error breakdown by source (e.g., transfer vs. flow estimation vs. data synthesis)?**
> The lower performance mainly comes from imperfect modality transfer. For example, the NIR modality has a much narrower spectral range and only one channel, while RGB covers a wider spectrum with three channels. This makes the transfer from NIR to thermal more difficult than the transfer from RGB, especially when perfectly aligned cross-modal pairs are not available for supervision. In Fig.11 of the revised manuscript, we present visual examples on RGB–T and NIR–T from the same scenes. Compared with the transferred images from RGB, the transferred images from NIR contain blurrier structures and less distinct edges in several regions, which lead to larger alignment errors. We have added the above discussion in the revised manuscript (lines 941-948).
>
> ---
>
> **References:**
> [A1] UniDepth: Universal Monocular Metric Depth Estimation. CVPR, 2024.
> [A2] Deep Depth Estimation from Thermal Image. CVPR, 2023.
> [A3] Deep Depth Estimation from Thermal Image: Dataset, Benchmark, and Challenges. arXiv, 2025.
> [A4] Depth Anything V2. NeurIPS, 2024.
> [A5] Metric3D v2: A Versatile Monocular Geometric Foundation Model for Zero-shot Metric Depth and Surface Normal Estimation. TPAMI, 2024.

---

> > ### Author Response · Authors · 2025-11-27
> >
> > Dear Reviewer,
> >
> > Thank you sincerely for your thoughtful and constructive comments.
> >
> > With the discussion period nearing its end, we would like to confirm whether our responses have fully addressed your concerns. Should any further clarification be needed, we would be happy to provide it promptly.
> >
> > We truly appreciate your time and insights, and welcome any additional questions you may have.
> >
> > Best regards,
> >
> > The DCFlow authors

---

### Author Response · Authors · 2025-11-19
**Response to all reviewers**

We thank all reviewers for their constructive feedback and for recognizing the strengths of our work, including:
- **Effective framework and designs.** Reviewers highlighted our design "clearly addresses a key limitation in present unsupervised cross-modal flow methods" (PNVY), "is well-suited for scenarios where cross-modal ground-truth data is scarce" (te33), and acknowledged the strengths of the decoupled optimization recipe, the geometry-aware synthesis pipeline with robust loss, and the cross-modal consistency constraint (NaNT).
- **Strong experimental results.** Reviewers highlighted that "the experimental section is thorough" (PNVY), "comprehensive ablation study is conducted" (te33), and "the presented results look good" (xLM8).
- **Well-structured paper.** Reviewers noted that "the paper is well-structured and easy to understand" (xLM8).

We summarize the updates made to the revised manuscript as follows:
- **Academic writing.** We have modified the reasoning of our design (NaNT, lines 160-192), the use of "unsupervised" (te33), the citation format (xLM8), the claim of EPE reduction (xLM8, lines 85-92), and the formatting of Eq.(12) (xLM8, lines 322-323).
- **More implementation details.** We have added the implementation details of the cross-modal consistency constraint (te33, lines 733-745).
- **Discussion of relevant works.** We have added the discussion of relevant works of data synthesis and outlier-robust loss (te33, lines 44-49, 135-137, 288-290).
- **Discussion and ablation of the robustness under imperfect monocular depth.** We have discussed the robustness of DCFlow under imperfect monocular depth, and added ablations on depth degradation and depth networks (PNVY, NaNT, lines 425-458).
- **More ablation studies.** We have added ablations on modality transfer direction (xLM8, lines 463-468), the outlier-robust loss (NaNT, lines 858-883), the affine ranges in the cross-modal consistency constraint (NaNT, lines 884-888), and the effectiveness on different flow networks (xLM8, lines 397-403).
- **More experimental results.** We have added the experimental analysis of efficiency, and generalization to unseen domains and sensors (NaNT, te33, lines 892-905, 960-963).
- **More qualitative analysis.** We have added the analysis of low-performing modalities (PNVY, lines 941-948), the training process of DCFlow (te33, xLM8, lines 949-955), and the valid mask for flow supervision (PNVY, lines 914-917).

The revisions are highlighted in **blue** in the updated version. Below we provide point-by-point responses to the individual comments of each reviewer. We look forward to further discussions and appreciate the opportunity to refine our work based on the reviewers’ feedback.

---

### Author Response · Authors · 2025-12-02
**Rebuttal Summary and Request for Consideration**

Dear Handling Area Chair,

We sincerely thank the anonymous reviewers for their time, thoughtful feedback, and voluntary service, and we greatly appreciate your coordination and support throughout the review process.

---

This paper was reviewed by four expert reviewers, with **consistently positive assessments (6, 6, 6, 6)**. Unfortunately, because the leaked information had already spread, the reviewers were unable to update their final scores after our rebuttal, despite our submission of detailed responses to all of their weaknesses and questions in the corresponding comment sections.

---

For your reference, we summarize below the main points addressed in our rebuttal, including our responses to the reviewers’ concerns and the corresponding clarifications and revisions. In the revised manuscript, the key changes are as follows:
 - We have added a discussion of the robustness of DCFlow under imperfect monocular depth, detailed ablations on depth degradation and different depth networks, and visual examples to illustrate the robustness of flow supervision. **[Lines 425-458, Lines 914-917, Table 2(c), Table 2(d), Fig. 10] @ Reviewer PNVY (W1, Q1), NaNT (W2, Q1)**
 - We have analyzed that the low performance of certain modalities is mainly caused by the difficulty of modality transfer. **[Lines 941-948, Fig. 11] @ Reviewer PNVY (Q2)**
 - We have revised the manuscript to better illustrate the reasoning behind our design. **[Lines 160-192] @ Reviewer NaNT (W1)**
 - We have added an ablation study on the value of τ in the outlier-robust loss and on the affine range of the cross-modal consistency constraint, and we have compared our outlier-robust loss with alternative robust loss functions. **[Lines 857-888, Table 5, Table 6, Table 7] @ Reviewer NaNT (W2, Q2)**
 - We have provided an analysis of efficiency and runtime for all comparison approaches, and added more quantitative experiments on generalization to unseen domains and sensors. **[Lines 892-905, Lines 960-963, Table 8, Table 9, Table 10] @ Reviewer NaNT (W3), te33 (W3)**
 - To improve clarity and correctness, we have revised the manuscript to refine the use of the term “unsupervised”, adjust the citation format, clarify the claim of EPE reduction, and correct the formatting of Eq. (12). **[Lines 33-36, Lines 74-77, Lines 85-92, Lines 322-323] @ Reviewer te33 (W1), xLM8 (W1, W2)**
 - We have discussed relevant works on data synthesis and outlier-robust loss, and clarified the contribution of our method. **[Lines 44-49, Lines 135-137, Lines 288-290] @ Reviewer te33 (W2)**
 - We have added more details of the training process of DCFlow, together with visualizations including loss curves and qualitative results at different training iterations. **[Lines 949-955, Fig. 12] @ Reviewer te33 (W4), xLM8 (Q2)**
 - We have added a detailed explanation of the data augmentation used in the cross-modal consistency constraint. **[Lines 733-745] @ Reviewer te33 (Q1)**
 - We have updated the ablation study on different flow networks by comparing them with the appearance-based optimization baseline. **[Table 2(e)] @ Reviewer xLM8 (W3)**
 - We have added an ablation study on the direction of modality transfer. **[Lines 463-468, Table 3] @ Reviewer xLM8 (Q1)**

---

Thank you very much for your time and consideration. We would greatly appreciate it if you could take into account the clarifications and progress made during the rebuttal when evaluating our submission.


Best regards,

The DCFlow authors

---

### Meta-Review · Area_Chair_y4y6 · 2026-01-06

**Summary:**

The paper proposes DCFlow, an unsupervised framework for cross-modal flow that trains a modality-transfer network and a flow network with a decoupled scheme plus a cross-modal consistency objective. The paper received initial scores of 6/6/6/6 and remained unchanged before the reset.

The reviewers recognized this work with effective framework and designs, thorough experimental results, and easy to understand. Their main concerns include more implementation details, ablation studies, experimental results, and qualitative analysis, which are well addressed in the response.

After a careful assessment of the submission, reviews, response, and the discussion, the AC recommends acceptance. The authors are encouraged to further refine the paper by incorporating the reviewers’ valuable suggestions.

**Reviewer Concerns:**

The reviewers recognized this work with effective framework and designs, thorough experimental results, and easy to understand. Their main concerns include more implementation details, ablation studies, experimental results, and qualitative analysis, which are well addressed in the response.

**Reviewer Scores:**

The manuscript received initial review scores of 6/6/6/6. After the rebuttal/discussion and before the reset, the score remained unchanged.

Since most concerns from the Reviewers are addressed after the rebuttal (see 'Reviewer Concerns'), I would approximate 6/6/6/6 as the final score.

---

### Decision · Program_Chairs · 2026-01-26

Accept (Poster)